# Drivers of extreme burnt area in Portugal: fire weather and vegetation

Tomás Calheiros[1], Akli Benali[2], Mário Pereira[3,4], João Silva[2], and João Nunes[1,5]

[1]cE3c: centre for Ecology, Evolution and Environmental changes, Faculdade de Ciências, Universidade de Lisboa, Lisboa, Portugal
[2]Forest Research Centre, School of Agriculture, University of Lisbon, Tapada da Ajuda, 1349-017 Lisboa, Portugal
[3]Centro de Investigação e de Tecnologias Agro-Ambientais e Biológicas (CITAB), Universidade de Trás-os-Montes e Alto Douro, Vila Real, Portugall
[4]IDL, Universidade de Lisboa, Lisboa, Portugall
[5]Soil Physics and Land Management Group, Wageningen University and Research, Wageningen, Netherlands

**Correspondence:** Tomás Calheiros (tlmenezes@fc.ul.pt)

**Abstract.** Fire weather indices are used to assess the effect of weather on wildfire behaviour and to support fire management. Previous studies identified the high Daily Severity Rating percentile (DSRp) as strongly related to the total burned area (BA) in Portugal, but it is still poorly understood how this knowledge can support fire management at a smaller spatial scale. The aims of this study were to 1) assess if the 90th DSRp (DSR90p) threshold is adequate to estimate most of BA in mainland Portugal; 2) analyse the spatial variability of the DSRp threshold that explains a large part of BA, at higher resolution; and, 3) analyse if vegetation cover can justify the DSRp spatial variability.

We used weather reanalysis data from ERA5-Land, wildfire and land use data from Portuguese land management departments for an extended summer period (15th May to 31st October) from 2001 to 2019. We computed and related DSRp with large wildfires (BA > 100 ha) and land use to clarify the effectiveness of the DSRp for estimating BA in Portugal and assess how vegetation influences it.

Results revealed that the DSR90p is an adequate indicator of extreme fire weather days and BA in Portugal. In addition, the spatial pattern of the DSRp associated with most of the total BA shows variability at the municipality scale. Municipalities, where large wildfires occur with more extreme weather conditions, have most of the burned areas in forests and are in coastal areas. In contrast, municipalities, where large wildfires occur with less extreme weather conditions, are predominantly covered by shrublands and are situated in eastern and inland regions. These findings are a novelty for fire science in Portugal and should be considered by fire managers and fire risk assessors.

## 1 Introduction

Wildfire incidence is defined as the number of fire events and/or burnt areas (BA) and strongly depends on the weather and climate, especially in regions with a Mediterranean type of climate, Csa and Csb in the Köppen-Geiger climate classification (Beck et al., 2018). This climate is characterized by mild and rainy winters and springs, favouring vegetation growth, while dry and hot summers promote thermal and hydric stress of live fuels and dryness of dead fuels (Romano and Ursino, 2020). In the

western Mediterranean, the influence of climate variability on wildfire incidence became increasingly evident after the 1970s, following a fire regime change, from fuel-limited to drought-driven (Pausas and Fernández-Muñoz, 2012). The main factor for this change was the increase in fuel load and continuity due to rural depopulation and land abandonment (Moreira et al., 2011; Moreno et al., 2014). These changes in landscape and population favoured the occurrence of large wildfires (Ferreira-Leite et al., 2016), which can also modify the landscape in the Mediterranean region (e.g. Stamou et al., 2016) influenced by regeneration patterns, topography and local fire histories. However, large wildfires tend to occur with severe fire weather conditions, being rare in other meteorological conditions (Telesca and Pereira, 2010).

Heatwaves and droughts have a strong influence on fire incidence, as shown by several studies in the last years in Mediterranean Europe (e.g., Duane and Brotons, 2018; Sutanto et al., 2020). The impacts of droughts on vegetation create favourable conditions for the ignition and spread of wildfires, especially during summer (Pausas and Fernández-Muñoz, 2012; Russo et al., 2017), but also in winter (Amraoui et al., 2015; Calheiros et al., 2020). In addition, fire incidence increased dramatically with the combined effect of prolonged drought and heatwaves on vegetation (water and heat stress), as pointed out by Ruffault et al., (2018). Wildfire incidence in Mediterranean Europe is expected to increase in the future because of climate change, especially due to global warming and changes in the precipitation regime (Sousa et al., 2015; Turco et al., 2018).

The Iberian Peninsula is the European region with the highest wildfire incidence which causes large property damages and fatalities (San-Miguel-Ayanz et al., 2020). In particular, Portugal has been severely affected by wildfires in the last decades, especially in 2003, 2005 and 2017, mainly as a consequence of anomalous atmospheric synoptic patterns and extreme weather conditions (Gouveia et al., 2012; Trigo et al., 2006; Turco et al., 2019). Other studies identified weather types, most of them connected with heatwaves or droughts in the western Iberian Peninsula, associated with the occurrence of large wildfires (Rodrigues et al., 2020; Vieira et al., 2020).

Fire weather danger indices are commonly used to assess the current and/or cumulative effect of atmospheric conditions on fuel moisture and fire behaviour. The Canadian Forest Fire Weather Index (FWI) System (CFFWIS) consists of six components that account for those effects (Van Wagner, 1987), including the Daily Severity Rating (DSR). The CFFWIS is extensively used in Portugal, both for research and operational purposes, having the best results when compared with other fire indices (Viegas et al., 1999; IPMA, 2022). The 90th percentile of the DSR (DSR90p) is often used as the threshold for severe fire weather that is associated with large fires (Bedia et al., 2012; Carvalho et al., 2008; Fernandes, 2019; Silva et al., 2019). More recently, the 95th percentile of DSR (DSR95p) was also identified as a good indicator of extreme fire weather and well related to the BA in the Iberian Peninsula (Calheiros et al., 2020; Calheiros et al., 2021).

Fire regime can be defined, in a strict sense, by the spatial and temporal patterns of wildfire characteristics (e.g. occurrence, frequency, size, seasonality, etc), as well as, in a broad sense, by vegetation characteristics, fire effects and fire weather in a given area or ecosystem, based on fire histories at individual sites over long periods, generally resulting from the cumulative interaction of fire, vegetation, climate, humans, and topography over time (Krebs et al., 2010; Whitlock et al., 2010; NCWG, 2011).

An essential element for fire incidence is the vegetation and land use type. There have been important changes in land use since the 1960s in Portugal which are related to wildfire occurrence. Arable cropland decreased from 40% to only 12% of the

total area in 2006, at the national level; and forests declined since the 1980s, as a result of forest fires, in Central Portugal (Jones et al., 2011). The contribution of landscape-level fuel connectivity to wildfire size was evident in the 1998 – 2008 period (Fernandes et al., 2016). The analysis of Corine Land Cover maps for 2000 and 2006 and EFFIS BA perimeters, from 2000 to 2013 in Portugal, revealed an increase in the area of shrublands and a decrease in forest areas, along with socioeconomic changes, impact the fire regime (Pereira et al., 2014; Parente and Pereira, 2016; Parente et al., 2018b). In Portugal, eucalyptus expansion has not modified the fire regime, but the rising undermanaged and abandoned forest plantations, especially after large-fire seasons, are a concern for the future (Fernandes et al., 2019).

Shrublands are more susceptible to wildfires, whereas agricultural areas and agroforestry systems are less likely to burn, as revealed by several studies (Carmo et al., 2011; Nunes, 2012; Meneses et al., 2018a). Barros and Pereira, (2014) identified shrublands as the most wildfire-prone land cover, followed by pine forests while, on the contrary, annual crops and evergreen oak woodlands tend to be avoided by wildfire. Ferreira-Leite et al., (2016) concluded that uncultivated land (shrublands, grasslands, and other sparse vegetation) was the most important factor affecting BA, considering large wildfires, greater than 100 ha. Topography and uncultivated land were significant factors determining BA, in a study for the 1980-2014 period conducted at the municipal level (Nunes et al., 2016). Additionally, there is evidence of an extending urban-rural interface in Portugal, due to an increase in the urban area since 1990, which contributes to an increase in fire incidence (Silva et al., 2019), especially in those regions (Tonini et al., 2018).

Land use interfaces, in particular those between forests and other land use types (shrublands, agricultural and urban areas), have a significant effect on human-caused wildfire occurrence in Mediterranean Europe, increasing fire risk due to human causes (Vilar et al., 2016). In the Iberian Peninsula, shrublands and pine forests have registered larger BA (Barros and Pereira, 2014; Pausas and Vallejo, 1999). This fact can be explained by the increasing landscape homogenization, due to shrublands expansion and agricultural abandonment, as observed by Lloret et al. (2002).

Wildfires in Portugal were the subject of several studies that developed zoning approaches to identify regions with similar fire regimes using solely BA data (Kanevski and Pereira, 2017; Scotto et al., 2014; Silva et al., 2019) or combined with fire weather indices (Calheiros et al., 2020, 2021; Jimenez-Ruano et al., 2018), large fire-weather typologies (Rodrigues et al., 2020), population density, topography, land cover changes (Oliveira et al., 2017) and net primary production (Fernandes, 2019). Their results indicate that Portugal can be divided into two (dividing the north and south of Tajo River) or three main clusters (the north part further divided in western and eastern). The spatial and temporal distribution of wildfires presents clustering patterns, suggesting that small fires are more dependent on local topographic or human conditions, while large fires are a consequence of infrequent causes or with shorter periods such as weather extreme events (Pereira et al., 2015). The temporal pattern is characterized by periodicities and scaling regimes (Telesca and Pereira, 2010) including a main summer fire season and a secondary spring peak, both driven by the type of climate and the occurrence of extreme weather conditions (Amraoui et al., 2015; Trigo et al., 2016; Calheiros et al., 2020).

A previous study assessed the recent evolution of spatial and temporal patterns of BA and fire weather risk in the Iberian Peninsula and concluded that the DSR95p is a good indicator of extreme fire weather and is well related to the BA, noticeable in the similar intra-annual variability pattern in four pyro-regions (Calheiros et al., 2020). This robust link was used to anticipate

fire regime changes caused by future climate change, revealing the potential displacement of fire regimes to the north (Calheiros et al., 2021). However, previous studies did not look at additional factors, for example: if the high percentiles of DSR are also linked with high values of BA in Portugal or Spain; if the high percentiles of DSR are similarly spatially distributed in all Iberia; or if, there is some spatial variability, what are the main reasons that can explain it. These knowledge gaps drove us to investigate if the DSRp value identified for the entire Iberian Peninsula is equally adequate to estimate BA in mainland Portugal, given its characteristics. Furthermore, we intended to study the variability of the relationship between DSRp and BA, together with the main factors of this variability. Accordingly, the objectives of this work were:

1) to assess if the DSR90p threshold is adequate to estimate most of BA in mainland Portugal;

2) to analyse the regional variations of the DSRp threshold that explains a large part of BA, at higher resolution, and;

3) to analyse if vegetation cover can explain the spatial variability of the DSRp.

In summary, this study aims to clarify the effectiveness of the DSRp for estimating BA in Portugal and how this relationship is influenced, namely by vegetation.

## 2    Data and methodology

### 2.1    Study Area: Portugal

This study focuses on mainland Portugal topographically characterized by mountainous ranges in the north and central regions and vast plains in the south, divided into 23 NUTS III regions which, in turn, are subdivided into 278 municipalities (Fig. 1). The BA variability is mainly influenced by the precipitation anomaly in spring and the occurrence of abnormal atmospheric patterns that generate very hot and dry days in the western Iberian Peninsula during summer (Pereira et al., 2005). Most (97%) of the total number of extreme wildfires (with BA $\geq$ 5000 ha) were active during heatwaves (Parente et al., 2018a) while almost (90%) of extreme wildfires during the 1981 – 2017 period occurred within a region affected by drought (Parente et al., 2019). The territory of Continental Portugal is mostly covered by forests (39%), agricultural lands (26%), shrublands (12%) and agroforestry systems (8%), according to data from *Direção Geral do Território* (DGT, 2019a). The most common tree species are *Eucalyptus Globulus* (26% of all forests), *Pinus Pinaster* (22%), both prevalent in the north and centre; and *Quercus suber* (22%), with larger areas in the south, using forest data from *Instituto Nacional da Conservação da Natureza e das Florestas* (ICNF, 2019). Pyro-regions shown in Fig. 1 are both characterized by a high peak of BA centred in August and a much smaller one in March. The main difference between the NW and SW pyro-region is the larger values of BA in the NW pyro-region, compared with the SW, especially in August (Calheiros et al., 2020).

### 2.2    Burnt Area

Wildfire data used in this study are burnt area polygons derived from satellite data provided for the 2001 – 2019 period by Portuguese national authorities (ICNF, 2020). This dataset was successfully used in many other studies, by a large number of authors for a wide variety of purposes (Bergonse et al., 2021; Tarín-Carrasco et al., 2021). Only wildfires with BA>100 ha that

occurred during the extended summer season, here defined between 15th May and 31st October, were considered in this study. It is important to explain these methodological options.

The focus on relatively large wildfires (here defined as wildfires with BA>100 ha) has two main reasons. First, mainland Portugal registers a huge number of small wildfires but they account only for a small amount of total BA (TBA). For example, wildfires with BA>100 ha are just about 1% of all wildfires but account for 75% of TBA (Pereira et al., 2011). Second, wildfires in Portugal are mainly (99.4%) caused by humans, either by negligence (about one-quarter of the total number of wildfires with known cause) or intentionally (about three-quarters), associated with the use of fire, accidents and structural/land use (Parente

et al., 2018b), which means that small wildfires can occur with relatively low DSR.

The study only considered wildfires that occurred during the 15th May – 31st October period because of also two main reasons: (i) BA caused by large wildfires within this period accounts for 97.5% of TBA; and, (ii) the secondary peak of fire incidence in Portugal occurs in late winter/early spring when DSR is lower and depends much more on drought than high air temperature (Amraoui et al., 2015; Calheiros, et al., 2020). The datasets and wildfire metrics used in this study are summarized in Table 1

and Table 2, respectively.

## 2.3    Meteorological Data and Fire Weather Indices

We used the DSR which is more accurate to rate the expected efforts required to suppress or control a wildfire, being an additional component of the FWI system (De Groot, 1987; Van Wagner, 1987).The indices of the FWI system were computed for the 2001 – 2019 study period with the equations provided by Van Wagner and Pickett (1975) and daily values at 12h00UTC

of air temperature and relative humidity (at 2 meters), wind speed (at 10 meters), and accumulated total precipitation (Table 1). Data on the meteorological variables were obtained from the latest (fifth) generation of European Centre for Medium-Range Weather Forecasts (ECMWF) atmospheric reanalyses of the global climate (ERA5-Land). The ERA5-Land dataset was loaded from the Copernicus Climate Change Service (C3S, 2020), with a much higher spatial resolution (0.1°lat x 0.1°long; the native resolution is 9 km) and temporal (hourly) resolution than the previous reanalysis data service, that was widely used and with

good performances for different purposes, including FWI calculation in Portugal (Bedia et al., 2012). ERA5 and ERA5-Land present agrees well with observations and present smaller biases than the other reanalysis datasets (Pelosi et al., 2020; Guo et al., 2021; Hassler and Lauer 2021; Nogueira, 2020), being recognized as the best or one of the best global atmospheric reanalysis datasets (Huai et al., 2021; Muñoz-Sabater et al., 2021; Urban et al., 2021) and used worldwide (Chinita et al., 2021; Sianturi et al., 2020; Sun et al., 2021; Araújo et al., 2022).

## 2.4    Vegetation and land use data

The land use and land cover (LULC) map for 2018 (COS2018) was provided by Portuguese national authorities (DGT, 2019a). This map has 83 land use or land cover classes clustered into 9 major classes: artificialized territories, agriculture, pastures, agroforestry, forests, shrublands, open spaces or areas with sparse vegetation, wetlands and bodies of water (Caetano, 2017). The burnable areas considered in this study were forests, shrublands, agriculture (which includes pastures and areas of agri-

155 cultural species highly susceptible to wildfires), agroforestry and other burnable areas (areas with sparse vegetation, peatlands

and coastal marshes). COS2018 was produced using aerial photographs from the same year, with more 35 classes than the previous versions. The dataset is available in shapefile format, composed of polygons with a spatial resolution of 1 hectare (DGT, 2019b). In particular, we organized the data into five major land-use types: forest, shrublands, agriculture, agroforestry and other burnable areas (Table 1). It is important to note that we did not aim to analyse any shrub- or forest-specific type.

## 2.5 Analysing burnt area and fire-weather relationship

The relationship between wildfires and weather was based on derived data, processed as described below. The starting and ending dates of each wildfire were fundamental to attributing the DSR to each BA. The dating process of the BA polygons relied on MODIS satellite data and the methodology of Benali et al. (2016). It was possible to estimate the starting and ending dates as well as ignition locations for 2016 wildfire events, corresponding to 92% of the initial total BA.

Daily DSR was computed for the study period and all ERA5-Land grid points within the territory of Continental Portugal. In the case of the analysis carried out for the entire mainland Portugal, the value of the DSRp associated with each wildfire was the maximum value of DSR registered in the area affected during the duration of the wildfire. When the analysis is carried out based on the municipalities, the procedure is similar with one exception: when a wildfire affected more than one municipality, the BA in each municipality was allocated to each municipality and analysed separately as a single wildfire event. The division of the BA between affected municipalities can introduce noise in the data since artificially generates BA which can be relatively small but associated with high or very high DSRp. To circumvent this potential problem, we decided to analyze BA percentages, relative to the total burnt area that occurred in a municipality during the study period, which reduces the influence of small wildfires on the final results.

We only selected (175) municipalities (from 278) affected by more than three wildfires and TBA > 500 ha. Restricting the analysis to the administrative units with sufficient data aims to increase the results' robustness and prevent potential interpretation errors. The selection of the maximum value of DSR to associate with wildfires is justified by the low spatial variability of the DSR, the small size of administrative units and the native reanalysis data resolution (C3S, 2020).

For the first objective, we start by making and analysing plots of BA metrics *vs.* DSRp (Table 2) for all the 2016 large wildfires that occurred in mainland Portugal during the study period, in the following order:

1) We first compared the BA values with DSRp and analysed them.

2) Those results lead us to sort BA data by the respective DSRp, compute accumulated values of BA, normalize it using the natural logarithm and plot against DSRp to assess if this relationship is linear.

3) Subsequently, we analysed if a fixed threshold of DSR for extreme days - DSR90p - is adequate to estimate extreme fire weather and is well related to large FTBA, for the entire territory. It is important to note that FTBA was calculated as the difference between 100 and the percentage of TBA corresponding to a certain DSRp (Table 2). This methodology was made with the purpose to visualize the TBA that burns above a DSRp threshold. We considered the correspondent 80% and 90% of FTBA as sufficient to classify DSRp as the extreme threshold, justified by the results of Pereira et al., (2005), which showed that 80% of TBA occurs in 10% of summer days.

## 2.6 Analysing clusters of burnt area

Potential clustering was assessed using the curves of FTBA *vs.* DSRp for all the selected municipalities. The high number (175) of these administrative regions difficult the interpretation of the results. Therefore, cluster analysis was performed to identify the major macro-scale spatial patterns and to objectively and statistically assess the significant differences between the results obtained for different municipalities.

The following notation was adopted to describe the linkages (the distance between two clusters) used in the *complete* clustering

method (The MathWorks Inc, 2021):

   • Cluster *r* is formed from two clusters.

   • $n_r$ is the number of objects in cluster *r*.

   • $x_r i$ is the *i*th object in cluster *r*.

   • *Complete linkage (d)*, also called the *farthest neighbour*, which uses the largest distance between

objects in the two clusters (Eq.1).

$$d(r,s) = max(dist(x_r i, x_s j)), i \in (1,\ldots,n_r), j \in (1,\ldots,n_s)$$  (1)

A distance metric is a function that defines the distance between two observations. The MATLAB function *pdist* used in this study can compute the pairwise distance between pairs of observations with different metrics. We applied the correlation distance because it provides a more easily interpretable dendrogram.

Given an *m*-by-*n* data matrix X, which is treated as *m* (1-by-*n*) row vectors *x1, x2, ..., xm*, in this case *m* being the number of analyzed municipalities and *n* the number of equidistant sampling points in the DSRp scale, the correlation distance between the vector xs and xt are defined in Eq.2:

$$d_{st} = 1 - \frac{(x_s - \overline{x_s})(x_t - \overline{x_t})'}{\sqrt{(x_s - \overline{x_s})(x_s - \overline{x_s})'}\sqrt{(x_t - \overline{x_t})(x_t - \overline{x_t})'}}$$  (2)

where $\overline{x_s}$ is described in Eq.3:

$$\overline{x_s} = \frac{1}{n}\sum_j x_{sj} \text{ and } \overline{x_t} = \frac{1}{n}\sum_j x_{tj}$$  (3)

The selected threshold $(1-R^2) = 0.35$ means that the coefficient of determination in the municipalities within the same cluster is higher than 0.65. This value was selected after a benchmarking analysis of the obtained dendrograms and results from an intended balance within the correlation between municipalities and the total number of clusters. For example, on one hand, if we had fixed 5 clusters, the correspondent correlation between municipalities within the same cluster will be only larger than

0.5, a value that we considered too low for this analysis. On the other hand, for a higher correlation, for example, 0.75, which corresponds to $(1 - R^2) = 0.25$, the number of clusters will be much higher, increasing the difficulty of interpreting the maps and dendrograms.

## 2.7 Analysing the influence of vegetation on the fire-weather relationship

The LULC was related to BA to accomplish the third objective of the study by computing several metrics (Table 2), namely: (i) the burnable area (BNA) in each municipality; (ii) the TBA in forests (BAF), shrublands (BAS), agriculture (BAA), agroforestry and other vegetation types; (iii) the ratio between forest and shrublands BNA (BNAF/BNAS) and TBA (TBAF/TBAS). Computations were made for each analysed municipality and cluster. Moreover, the spatial distribution of prevailing land-use types that were most affected by wildfires was investigated to identify which municipalities have a BA in forests larger than 50% or BA in shrublands larger than 40% of TBA. The adoption of different thresholds for BA in forests and shrublands is due to a much lower area of shrublands (12%) than of forests (39%) in continental Portugal (DGT, 2019a).

A contingency table, accuracy metrics and statistical measures of association were used to analyse the influence of the type of vegetation cover on the relationship between DSRp and TBA. The contingency table contains the number of municipalities that belong to a different group of clusters, i.e., different DSRp thresholds at 90% of TBA (DSRp90TBA) and is characterized by BAF > 50% or BAS + BAA > 40%. The objective was to relate the municipalities (within groups of clusters) with TBA in diverse vegetation cover types. Statistical measures of association were used for classification accuracy against a reference as, for example, municipalities with higher DSRp90TBA will have the largest TBA in forested areas, compared with other land use types; and accuracy metrics were computed according to this initial classification.

The list of accuracy metrics includes: (i) the Overall Accuracy (OA), which represents the samples that were correctly classified and are the diagonal elements in the contingency table, from top-left to bottom-right (Alberg et al., 2004); (ii) the User's Accuracy (UA), or reliability, that is indicative of the probability of a sample that was classified in one category belongs to that category; and, (iii) the Producer's Accuracy (PA), represents the probability of a sample being correctly classified (Congalton, 2001). Statistical measures are: the Chi-squared ($\chi$2) test (Greenwood and Nikulin, 1996), which tests the independence of two categorical variables; the Phi-test ($\Phi$) or phi coefficient (David and Cramer, 1947) is related to the chi-squared statistic for a 2×2 contingency table, and the two variables are associated if $\Phi$>0. Lastly, we computed the Cohen's Kappa coefficient, firstly presented by Cohen (1960) and recently analysed by McHugh (2012), which measures the interrater agreement of the two nominal variables. This coefficient ranges from -1 to 1 and is interpreted as < 0 indicating no agreement to 1 as almost perfect agreement.

## 3 Results

### 3.1 Burnt area and fire-weather relationship, at the national and municipality level

The scatter plot of BA as a function of DSRp (Fig. 2) reveals that most large wildfires, including those with the highest amounts of BA, were registered with the highest values of DSRp. For low DSR values, e.g. below the 80th percentile, the vast majority of BA is the lowest in the 2016 sample values. In addition, the scatter plot of the natural logarithm of the accumulated BA versus DSRp (Fig. 3) presents a linear relationship, with a very high coefficient of determination (R2=0.94) and p-value lower than the significance level. Furthermore, the logarithm of accumulated BA increases exponentially ($R^2 = 0.92$) for DSRp extreme

values (DSR>DSR90p), meaning that BA rises suddenly with extreme meteorological conditions. In summary, the results of these analyses reveal that: (i) wildfires can occur with a large spectrum of DSRp values, during the extended summer period; and, (ii) very large wildfires only occur with high DSRp.

The analysis of the dependence of FTBA with DSRp in the entire mainland Portugal territory (Fig. 4) revealed that most of the TBA occurred with very high DSRp values. For example, for days with DSR>50th DSRp (DSR50p) the FTBA is almost

255 100%, meaning that fires on days with lower DSR have a negligible impact on TBA (please see Section 2.5). Fires in days with DSRp between 85 and 95 were responsible for more than 80% of TBA in the 2001 – 2019 period, making this a good DSRp threshold for extreme days. This result justifies using the DSR90p at the national scale, which is widely used for a threshold of extreme values (Bedia et al., 2012; Carvalho et al., 2008; Fernandes, 2019; Silva et al., 2019).

However, if the analysis is performed at a higher spatial resolution, namely at the municipality level, some differences be-

260 come apparent (Fig. 5). The spatial distribution of DSRp for FTBA=80% (DSRp80TBA) or FTBA=90% (DSRp90TBA) in each municipality presents important differences between regions, together with more visible contrasts in DSRp90TBA than in DSRp80TBA. The much lower values of DSRp in the north-eastern (*Alto Tâmega*, *Terras de Trás-os-Montes*, *Douro* and northern *Beiras e Serra da Estrela*) and in the southern interior regions (*Alentejo Central* and *Baixo Alentejo*) should be highlighted. DSRp90TBA is higher in most of the coastal and some central hinterland municipalities (portions of *Área Metropolitana do*

*Porto*, *Viseu Dão-Lafões*, *Região de Coimbra*, *Beira Baixa* and *Região de Leiria*), reaching values similar to the mean country-level value (85<DSRp90TBA<95). In some municipalities of the northern and central hinterland, DSRp90TBA is between 60 and 70, particularly in *Douro* and *Terras de Trás-os-Montes*. It is important to underline that DSRp80TBA>DSRp90TBA which is a consequence of the adopted methodology to perform this analysis (please see Section 2.5). This also helps understand why DSRp=50 is associated with FTBA=100% (Fig. 4). The spatial distribution of DSRp80TBA and DSRp90TBA suggests the

existence of clustering.

## 3.2 Burnt area clusters

The spatial distribution of DSRp80TBA and DSRp90TBA suggests the existence of clustering, which should also help explaining the feature similarities or differences between municipalities. Therefore, the municipalities were grouped into ten clusters based on the relationship between TBA and DSRp. Results disclose that cluster 10 is composed of just one municipality and,

consequently, was excluded from the dendrogram (Fig. 6) and further analysis.

The spatial pattern of Fig. 7 reveals a relatively homogeneous distribution of the municipalities of equivalent clusters and patches of municipalities belonging to consecutive clusters, meaning that municipalities with similar DSRp are often neighbours.

The FTBA *vs.* DSRp plots were produced for each cluster to illustrate and interpret the clustering results (Fig. 8). FTBA=100%

occurs for DSR90p in cluster 1, confirming that large wildfires in these municipalities only occurred with very extreme meteorological conditions. The FTBA *vs.* DSRp curves for the first three clusters present a very steep slope for the highest DSRp values, revealing that large wildfires take place in the municipalities of these clusters on days with high DSRp (above 90). Moreover, the FTBA *vs.* DSRp plots for these clusters present very low dispersion suggesting that the curves for the municipal-

ities of each of these clusters are very similar. These municipalities are located in the north and central western coastal areas, and also include mountain ranges (predominantly in *Alto Minho*, *Cávado*, *Área Metropolitana do Porto*, *Tâmega e Sousa*, *Região de Aveiro*, *Região de Coimbra* and *Alentejo Litoral*), within some central and south hinterland regions (parts of *Viseu Dão-Lafões*, *Beiras e Serra da Estrela*, *Médio-Tejo* and *Alto Alentejo*) and in the south coast (almost all of *Algarve*).

Clusters 4, 5 and 6 are prone to burn with less extreme conditions, where the median of DSR90p corresponds to 85 – 90% of TBA. The slope of FTBA *vs.* DSRp curves is less steep but the dispersion is higher than in the previous clusters, meaning that large wildfires can occur with lower values of DSRp. Both features suggest that in these clusters, wildfires tend to occur in a wider range of meteorological conditions. These clusters are spread throughout the country and can be viewed as a transition between the group of clusters with extreme (1, 2 and 3) and less extreme (7, 8 and 9) DSRp80TBA or DSRp90TBA.

Clusters 7, 8 and 9 can be considered as the group of lower DSRp clusters, due to the relatively lower values of the DSRp80TBA or DSRp90TBA, which range from 70 to 80%. Higher dispersion is also apparent, especially in cluster 9, which integrates municipalities where large wildfires can occur with lower values of DSRp (in some cases, below DSR50p). In this group of clusters, the slope of the FTBA *vs.* DSRp curves, at higher values of DSRp, is the lowest, especially in clusters 8 and 9. Nevertheless, the median curve of cluster 8 has a different behaviour, compared to the other two clusters: the steeper interval is between 70th and 80th percentile, meaning that a larger amount of BA occurs in less extreme conditions. The municipalities within these clusters are mostly located in the northern and central hinterland, particularly in *Alto-Tâmega*, *Terras de Trás-os-Montes*, *Douro*, *Beiras e Serra da Estrela* and *Beira Baixa*. Additionally, a few municipalities within these clusters belong to *Alentejo Central* and *Baixo Alentejo*, two provinces with a scarce number of fires and BA.

Box-plots of the DSRp80TBA and DSRp90TBA for the municipalities of each cluster (Fig. 9) are consistent with the previous results. Dispersion is considerably much higher in the latter than in the former case, especially in clusters 3, 7 and 8. In some municipalities of clusters 7 and 8, large wildfires, with the ability to exceed FTBA=10% (Fig. 8), start to occur with relatively low values of DSRp. Another notable difference is the boxplot medians: for DSRp90TBA they decrease with the ascending number of clusters as expectable, but not for DSRp80TBA, where they increase between clusters 4 and 5, between 6 and 7, and between 8 and 9.

## 3.3  Influence of vegetation on the fire-weather relationship

Therefore, we explored other features of the fire regime in mainland Portugal, namely BA metrics (Table 2), linked with vegetation, that could explain the similarities and differences observed in their patterns at the municipality level. The BNA and the BNAF/BNAS and TBAF/TBAS ratios in each municipality were assessed and analysed (Fig. 10). Additionally, the number of wildfires in each municipality was also evaluated (see Appendix).

The BNA (Fig. 10a) is much lower in coastal municipalities (except in *Algarve*) and in most of the northern and central hinterland, particularly in *Terras de Trás-os-Montes*, *Douro* and portions of *Beiras e Serra da Estrela*. These relatively low values are explained by the high population density and urban areas near the coastline or by agricultural patches in the countryside. On the other hand, higher BNA is found in the mountain ranges, especially in the northwest (some municipalities located in *Alto Minho*, *Cávado* and *Alto Tâmega*) as well as in some specific highly forested regions in the central hinterland (within

*Área Metropolitana do Porto*, *Viseu Dão-Lafões*, *Região de Coimbra*, *Região de Leiria*, *Médio Tejo* and *Beira Baixa*) and one municipality in *Algarve*. These patterns are justified by low population density, low availability of land suitable for agriculture, and, in some regions, extensive forest plantations.

The BNAF/BNAS (Fig. 10b) show that forest cover is prevalent in most of the analysed municipalities, especially near the west coast. Conversely, shrublands BNA is only dominant in a few municipalities located in the northern hinterland, particularly in *Alto Minho*, *Alto Tâmega*, *Douro* and *Beiras e Serra da Estrela*. However, the spatial distribution of the TBAF/TBAS (Fig. 10c) present some considerable differences, namely an extensive number of municipalities in the north, coastal and inland, that have larger TBA in shrublands, namely a large number of municipalities located in *Alto Tâmega*, *Tâmega e Sousa*, *Douro*, *Viseu Dão-Lafões* and *Beiras e Serra da Estrela*). Nevertheless, the municipalities with higher BNAF/BNAS correspond with those with larger TBAF/TBAS. Results of both maps are similar when analysing the southern provinces of the country (*Alto Alentejo*, *Alentejo Central*, *Alentejo Litoral*, *Baixo Alentejo* and *Algarve*), where almost all municipalities are characterized by higher forest BNA and TBA.

The spatial distribution of the clusters resembles the general pattern of LULC in Portugal (Fig. 11, bottom panel). In general, municipalities with high DSRp90TBA are located in regions of forests while municipalities with lower DSRp90TBA are located in regions where shrublands tend to be predominant. Analysis of BA in LULC type, made for each cluster, indicates that BA in forests (BAF) is notably higher than in shrublands (BAS), for the first five clusters than for the last four clusters (Fig. 11, top panel). This means that BAF is higher for clusters with higher DSRp90TBA while BAS is higher for clusters with lower DSRp90TBA. In addition, there is an increase in the fraction of BA in agricultural land associated with the decrease of DSRp90TBA. This amount is higher and about 10% – 20% in clusters 6 – 9, but lower in clusters 1 – 5.

Results show marked pieces of evidence between most coastal and northern/north-eastern hinterland municipalities, which present similar DSRp90TBA and, therefore, similar cluster distribution. Highest BAF characterizes the majority of the municipalities with the observed highest DSRp at 90% of TBA (generally above 85) while the territory with higher BAS is also characterized by lower DSRp90TBA (below 85). These clusters (7-9) also present relatively high percentages of BA in agriculture (mostly between 10 and 20%). It is also worth mentioning that some municipalities present similar BAF and BAS, although being located in the coastal regions, usually characterized by higher forest cover.

The land cover also helps to understand the DSRp80TBA and DSRp90TBA boxplots for each cluster, especially the higher dispersion in the latter in comparison with the former (Fig. 10). These dissimilarities are especially evident in cluster 8, which is the cluster with the highest BAS and BAA (twice the value of clusters 1 – 5) and less BAF (half the value of clusters 1 – 5). Additionally, cluster 8 is the one with a less BNA (not shown).

The combination of these factors could explain the high dispersion: high BAS can occur with low DSRp, high BAA is much more likely to occur with high DSRp; and, finally, low BNA prevents very large wildfires to occur, even with extreme DSRp.

A contingency table permitted objectively and quantitatively assess the influence of vegetation cover in the spatial distribution of the clusters and, therefore, also in DSRp90TBA. Table 3 is based on the results illustrated in Fig. 11 and aims to assess if the differences in groups of clusters or DSRp90TBA can be explained by the BA prevailing in forested areas or shrub-

land+agricultural zones. Specifically, it purposes to assess if municipalities of clusters 1 – 5, with DSRp90TBA>90, have higher BAF (BAF>50%), and, on the contrary, clusters 7 – 9, with DSRp90TBA<90, present higher BAS+BAA (BAS+BAA>50%). Results reveal that the number of municipalities in clusters 1-5 and BAF>50% is 4.6 times higher than the number of mu-
nicipalities in clusters 7-9 and BAF>50%. However, the number of municipalities of clusters 7-9 and BAS+BAA>50% is 1.3 higher than the number of municipalities of clusters 1-5 and BAS+BAA>50%. Consequently, the OA (71%), UA (71% – 70%) and PA (82% – 55%) reveal moderate to high accuracy. The BAS+BAA>50% threshold is probably a too demanding criterion for the DSRp90TBA=90 limit, as shrublands and agricultural land cover will also burn with higher DSRp in a large number of municipalities. For forests (BAF>50%), the accuracy is better, i.e., this threshold has been accurate in more than four times of
the municipalities that were incorrectly classified. Cohen's Kappa test allows us to conclude a fair agreement ($\kappa$=0.3828) and rejects the null hypothesis: observed agreement is not accidental (Landis and Koch, 1977). The $\Phi$ and C tests also corroborated that these variables are dependent, with similar values, 0.39 and 0.36, meaning moderate correlation (Frey, 2018) and the existence of a relationship (De Espindola et al., 2009), respectively. However, the $\chi2$ test results indicate that we can claim that the samples are independent (Frey, 2018), with an error risk of about 4e-06.
Thus, three out of four computed statistics prove a dependent relationship and, consequently, we can state that the cluster's spatial distribution patterns are correlated with vegetation type.

## 4  Discussion

### 4.1  Burned area and fire-weather relationship

The scatter plot of BA *vs* DSR indicates that BA strongly depends on DSR (Fig. 2). On one hand, large wildfires can occur on
370  days with a wide range of relatively low values of DSRp (DSRp<80) due to several reasons including rapid fire-suppression activities (e.g., firefighting) or fuel constraints (e.g., fuel breaks, geographical and landscape features). On the other hand, extremely large wildfires only occur on days of extreme fire weather as pointed out by several studies (Fernandes et al., 2016). According to our results only 6% of the TBA occurs with DSRp<80 and 12% of TBA are registered in wildfires with DSRp<90. The scatter plots of Log (accumulated BA) and FTBA *vs.* DSRp (Fig. 3 and Fig. 4) suggest that DSR90p is a suitable threshold
to identify extreme weather associated with high TBA, for mainland Portugal, which is in line with previous studies (Bedia et al., 2012; Carvalho et al., 2008; Fernandes, 2019; Silva et al., 2019).
However, analysis performed at a finer spatial scale (Fig. 5) discloses interesting deviations, namely differences between coastal areas and the hinterland municipalities. Large wildfires/high BA can occur in most of the inland municipalities in the northeast and parts of southern Portugal with DSRp<80, but can only occur in coastal and some mountainous municipalities with higher
DSR (DSR>DSR90p).
The cluster analysis based on the DSRp *vs* FTBA curves aimed to find groups of municipalities with similar fire-weather relationships. As expected, the spatial distribution of the clusters (Fig. 7) is also very similar to the DSRp80TBA and DSRp90TBA maps (Fig. 5), especially the marked differences between the coastal and hinterland municipalities of the northeast and south-central.

The curves of DSRp *vs* FTBA for the clusters (Fig. 8) show decreasing slopes and increasing variability with the decrease in the DSR, which means a trend for large wildfires to occur with less extreme weather conditions and greater variability between the municipalities of each cluster.

## 4.2 Influence of vegetation on the burnt area and fire-weather relationship

Differences in DSRp throughout the territory are expected due to distinct characteristic factors, including climate and land-
390 scape features. Mainland Portugal has two slightly different types of temperate (group C) climate, namely Csb (dry and warm summer) in the north and Csa (dry and hot summer) in the south, which promote different fire regimes in these two regions (Parente et al., 2016). LULC is also an important wildfire factor in Portugal (Barros and Pereira, 2014; Leuenberger et al., 2018; Parente and Pereira, 2016; Pereira et al., 2014; Tonini et al., 2018). Therefore, it is not surprising a high similarity between the spatial patterns of DSRp80TBA or DSRp90TBA and the LULC maps for Portugal (e.g., please see Fig. 4 of Parente and
395 Pereira (2016)). Other wildfire-related vegetation features were assessed (Fig. 10) to explain the heterogeneity of DSRp80TBA and DSRp90TBA maps (Fig. 5). The BNAF/BNAS ratio pattern shows higher BNA in forests in most of the territory but the TBAF/TBAS ratio reveals higher TBA in shrublands, especially in regions of lower DSRp80TBA and DSRp90TBA. These findings are in line with the higher land cover proneness to wildfires for shrublands and pine forests than for annual crops, mixed forests and evergreen oak woodlands (Barros and Pereira, 2014; Pereira et al., 2014).
Contingency tables, accuracy and statistical tests led us to conclude that vegetation types, particularly forest and shrublands, influence the spatial distribution of DSRp observed in Portugal.

The different vegetation cover can explain the spatial distribution of DSRp within mainland Portugal and, therefore, clusters' dissimilarities (Fig. 11). On one hand, extreme DSR extremes are strongly influenced by long-lasting severe droughts (not only during but before the fire season), heatwaves (during fire season) or both. Heat waves and droughts are important extreme
weather/climate events, promoting wildfire occurrence and spread, and, therefore, high BA (Russo et al., 2017; Parente et al., 2018a; Parente et al., 2019). On the other hand, shrublands are more likely to suffer from droughts than forests. As observed by Gouveia et al, (2012), during drought shrublands presented higher levels of dryness, whereas broad-leaved forests exhibited lower water stress. Coniferous forests are more resistant to short-term droughts than broad-leaved forests, because of their decreased vulnerability to xylem cavitation (Allen et al., 2010). Consequently, forests tend to burn only under extreme DSR
values, typically caused by simultaneous drought and heatwave, while shrublands (and also agricultural areas) can burn with lower DSRp. These facts can be additionally justified by biological features. In the Mediterranean region, precipitation is the main constraint to photosynthesis and growth (Pereira et al., 2007). This is particularly critical for shallow-rooted species, like those of the herbaceous vegetation and some shrub species, which are unable to access groundwater. It is less critical for deeply rooted species such as cork oak, and other drought-resistant Mediterranean species (Cerasoli et al., 2016).

## 4.3 Considerations and implications for management

LULC data can affect the relationship between extreme fire weather and BA. LULC changed during the 19 years (2001 – 2019) of the study period in many locations, including in the BA polygons. Effectively, Meneses et al., (2018b) observed that the main

land-use changes, for the 1990 – 2012 period, are related to reductions in forests and agricultural areas, together with increases in urban areas, with relatively small changes between 2000 – 2006 and 2006 – 2012 periods. Therefore, LULC changes do not significantly affect the findings, knowing that we only use LULC data for one year/inventory to assess wildfire selectivity. Understory vegetation can also be an important factor in fire vulnerability, spread and intensity (Espinosa et al., 2019; Fonseca and Duarte, 2017). Consequently, wildfires only tend to occur and spread in managed forests with very high DSR, higher than in unmanaged forests (Fernandes et al., 2019). However, land use data does not include forest management information. Despite the small fraction of managed forested areas, which Beighley and Hyde, (2018) roughly estimated as 20%, the lack of this information can influence our results, particularly in the municipalities with a significant share of managed forest areas.

It is also important to underline that, to identify the drivers of extreme BA in Portugal, we used objective methods and adequate statistics that ensure the robustness and statistical significance of the results. The description of the study carried out also includes the chronology of the performed analysis. In a previous study (Calheiros et al., 2020), the relationship between fire weather and fire incidence was analysed in-depth for the entire Iberian Peninsula. Among other results, they found that the DSR90p is a good indicator of extreme fire weather and is well related to the BA in the Iberian Peninsula. In this study, we started by verifying whether the relationship between DSRp and BA found, in general terms, for the Iberian Peninsula, was also verified in mainland Portugal, at the municipality level, and what is the spatial variability of the extreme value of DSRp above which most of the burned area is registered. To objectively interpret the obtained spatial patterns (Fig. 5), we complement and deepened the analysis with the use of clustering algorithms, to classify the municipalities into statistically different groups in terms of the relationship between FTBA and DSRp. The emerging patterns showed that all of those most likely factors, such as topography, altitude (Fig. 1), slope (please see Fig. 5 of Parente and Pereira, 2016), population density (please see Fig. 2 of Pereira et al., 2011), rural and urban area type (please see Fig. 3 of Pereira et al., 2011), road density/distance to the nearest road (please see Fig. 2a of Parente et al., 2018b) and climate type (please see Fig. 1a of Parente et al., 2016) were not able to explain the obtained spatial patterns. The only factor with a similar spatial pattern was the LULC, which is the reason why we decide to explore this possibility more deeply, with contingency tables and several accuracy metrics to assess the influence of the type of vegetation cover on the relationship between DSRp and TBA.

Finally, the results of this study are be a valuable resource in an innovative risk assessment system, improving the current wildfire risk mapping, taking into consideration the role of vegetation in the relationship between extreme weather and large wildfires. These maps are useful for forest management, landscape or land-use planning, firefighting, civil protection and other stakeholders. Our findings are innovative for fire science in Portugal, showing an important relationship between fire weather, wildfires and vegetation.

## 5    Conclusions

This work disclosed that the 90th percentile of DSR, used to identify extreme fire weather days, is a good indicator for the extreme BA in mainland Portugal. However, at higher resolutions, this threshold presents regional variations that should be considered, namely for landscape and wildfire management.

This analysis of the relationship between extreme fire weather (specifically DSRp) and fire incidence (specifically BA) leads us to conclude that LULC – a structural factor – influences the impacts of meteorological conditions – a conjectural factor of fire risk. To our knowledge, this is the first study that identifies and establishes that the relationship between fire weather and fire incidence depends on LULC, in Portugal.

The role of vegetation cover on these regional variations is an important outlook of our results. Shrublands are more suitable to burn in less extreme conditions than forests. Climate type and vegetation cover explain the DSRp spatial distribution dissimilarities, highlighting that landscape and forest management are key factors for the adaptation to future climate change, due to the highly probable vegetation changes.

To the best of our knowledge, the findings of this study have never been previously reported or published. These innovative
findings significantly contribute to forest and fire management because they identify the fire weather risk limits for the large wildfire occurrence in different vegetation types. The relationship between fire weather, fire incidence and vegetation became better known. The spatial variability of DSRp and its dependence on vegetation type has high operational value and should be considered by fire managers and risk assessors to help firefighters and civil protection in fire prevention and combat planning.

*Data availability.* This research was developed using three public data sources. The meteorological variables were obtained from the fifth
generation of ECMWF atmospheric reanalyses of the global climate (ERA5-Land) dataset (Copernicus Climate Change Service (C3S), 2017). Land use and land cover data were provided by Portuguese national authorities, respectively,*Direção Geral do Território* (DGT, 2019a), and the wildfire database from the *Instituto Nacional da Conservação da Natureza e das Florestas* (ICNF, 2020).

## 1 APPENDIX

In this section, we present the results that were important but not fundamental for this manuscript. The number of fires in
Portugal (Figure A1), in each analysed municipality, was assessed. The distribution of the number of wildfires, between 2001 and 2019, discloses a notable contrast between northern and southern provinces (the last ones considered as *Alto Alentejo*, *Alentejo Central*, *Alentejo Litoral*, *Baixo Alentejo* and *Algarve*). Wildfires were more frequent in the extreme northwest (*Alto Minho* and *Alto Tâmega*) and some municipalities located in *Beiras e Serra da Estrela*. Wildfire frequency is much lower in the south and on most of the western coast.

*Author contributions.* TC developed the code to analyse the data, produced the results and plots, and wrote the original draft of the manuscript. AB contributed to the supervision, the code to analyse data and produce plots, and also to the writing. MP contributed to the supervision, production of plots and writing. JNS contributed to the supervision, methodology and writing. JPN contributed to the supervision and writing. All authors contributed to the conceptualization and methodology of this research.

*Competing interests.* The authors declare that they have no conflict of interest.

*Acknowledgements.* This work was funded by the Portuguese Fundação para a Ciência e a Tecnologia through the PhD fellowship attributed to T Calheiros (PD/BD/128173/2016). Additional funding was obtained through the individual research grant attributed to JP Nunes (IF/00586/2015), research project FRISCO (PCIF/MPG/0044/2018), and research unit funding attributed to the CE3C, CITAB and Forest Research Centre (CEF) research centres (UIDB/00329/2020,UIDB/04033/2020 and IDB/00239/2020, respectively).

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

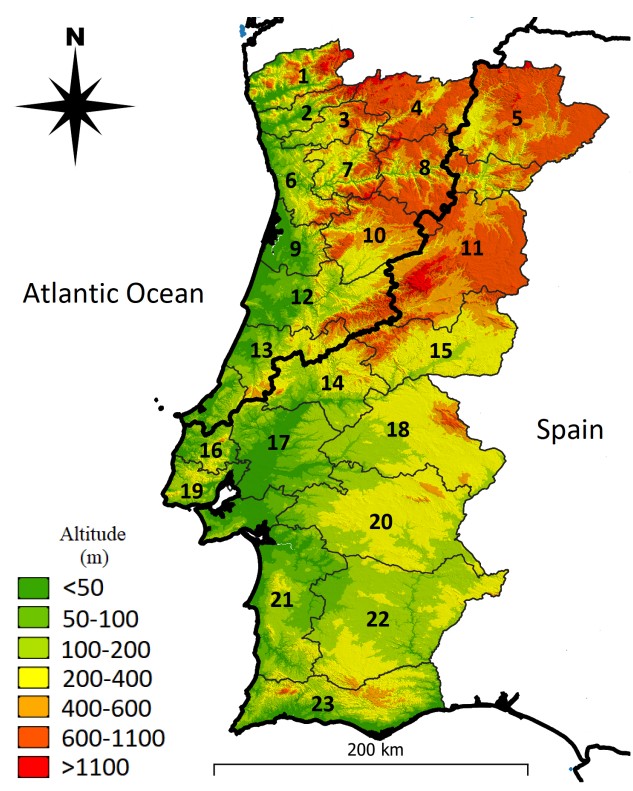

**Figure 1.** Mainland Portugal topography and administrative division based on NUTSIII provinces: *Alto Minho*(1), *Cávado*(2), *Ave*(3), *Alto Tâmega*(4), *Terras de Trás-os-Montes*(5), *Área Metropolitana do Porto*(6), *Tâmega e Sousa*(7), *Douro*(8), *Região de Aveiro*(9), *Viseu Dão-Lafões*(10), *Beiras e Serra da Estrela*(11), *Região de Coimbra*(12), *Região de Leiria*(13), *Médio-Tejo*(14), *Beira Baixa*(15), *Oeste*(16), *Lezíria do Tejo*(17), *Alto Alentejo*(18), *Área Metropolitana de Lisboa*(19), *Alentejo Central*(20), *Alentejo Litoral*(21), *Baixo Alentejo*(22) and *Algarve*(23). NUTSIII frontiers were loaded from the European Environment Agency (EEA, 2021) and altitude data from *Direção Geral do Território* (DGT, 2010). For comparison purposes, the borders (thick black line) of the pyro-regions found by Calheiros et al., (2020), were also added: the NW pyro-region is located in northwestern Portugal and the SW pyro-region in southwestern and eastern of the country.

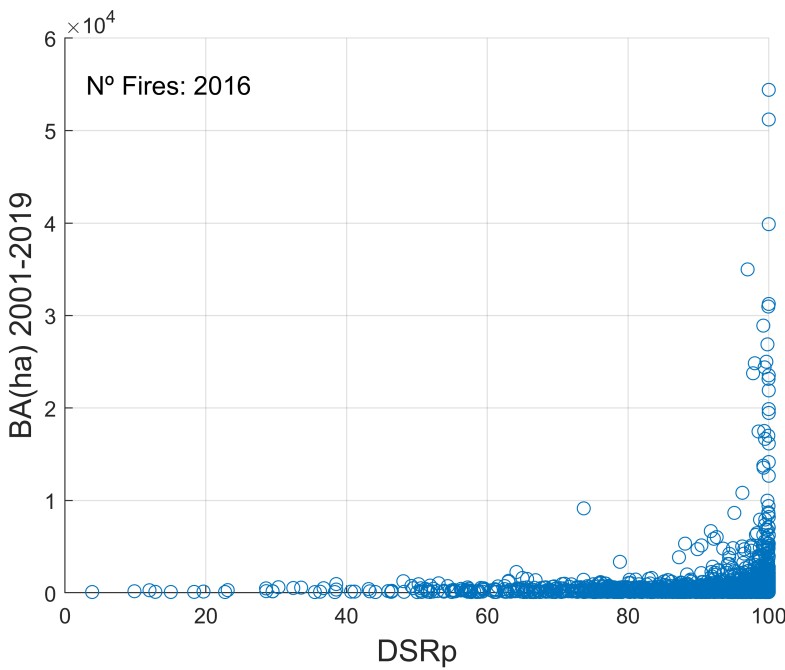

**Figure 2.** Scatterplot of the burnt area (BA) *vs.* DSR percentile (DSRp) for wildfires (blue circles) with BA>100 ha that occurred between May 15 and October 31, in the 2001 – 2019 period.

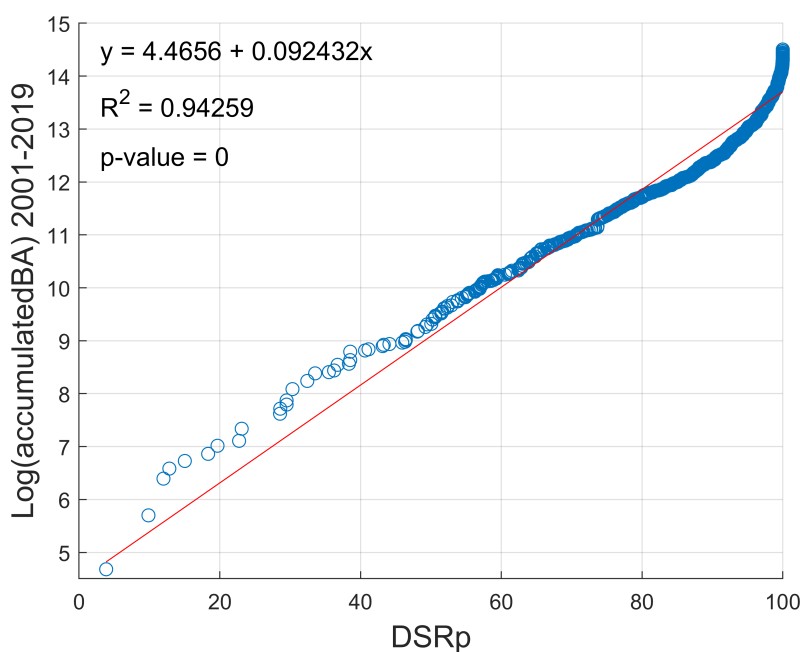

**Figure 3.** Scatterplot of the decimal logarithm of the accumulated burnt area (Log(accumulatedBA)) *vs.* DSR percentile (DSRp), considering the fires with an area larger than 100 ha that occurred between May 15 and October 31, in the 2001 – 2019 period. The blue circles represent each wildfire, with respective accumulated BA, after being sorted by the assigned DSRp. Best fit (red line), respective equation, $R^2$ and p-value are also presented.

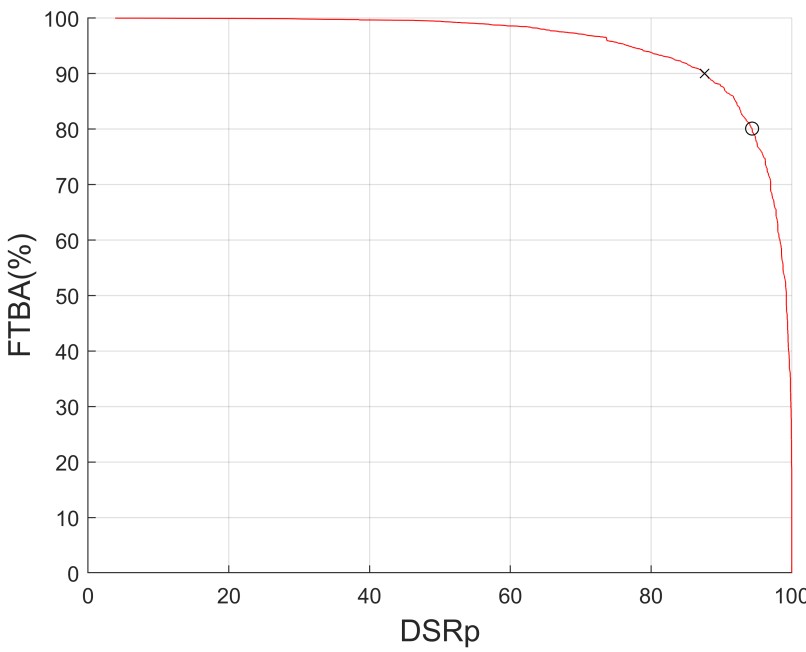

**Figure 4.** Fraction of total burnt area (FTBA) *vs.* DSR percentile (DSRp), computed for mainland Portugal, in the 2001 – 2019 period. The circle (cross) is the DSRp when the FTBA reaches 80% (90%).

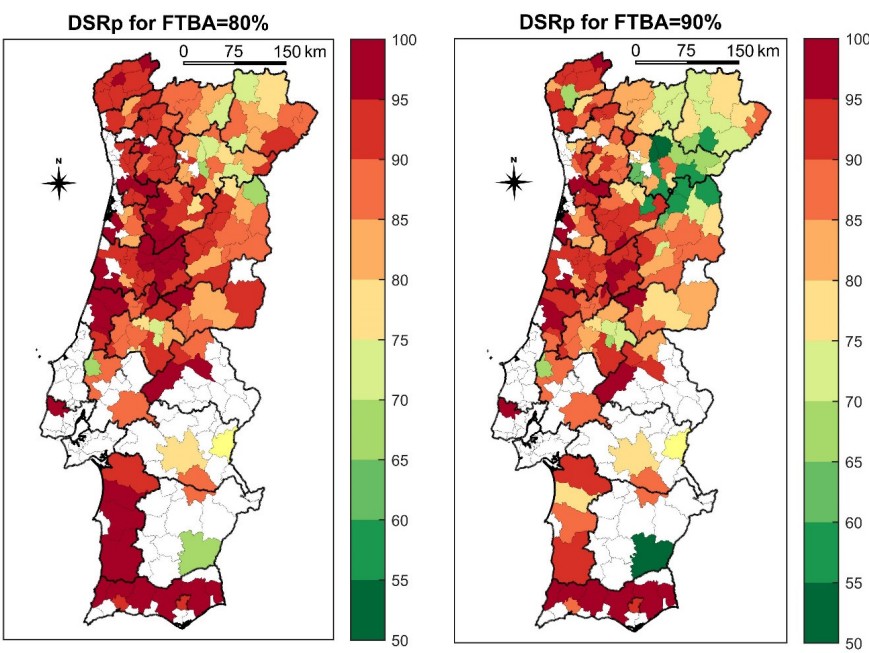

**Figure 5.** DSR percentile (DSRp) for 80% (left panel) and 90% (right panel) of the fraction of total burnt area (FTBA) in each municipality.

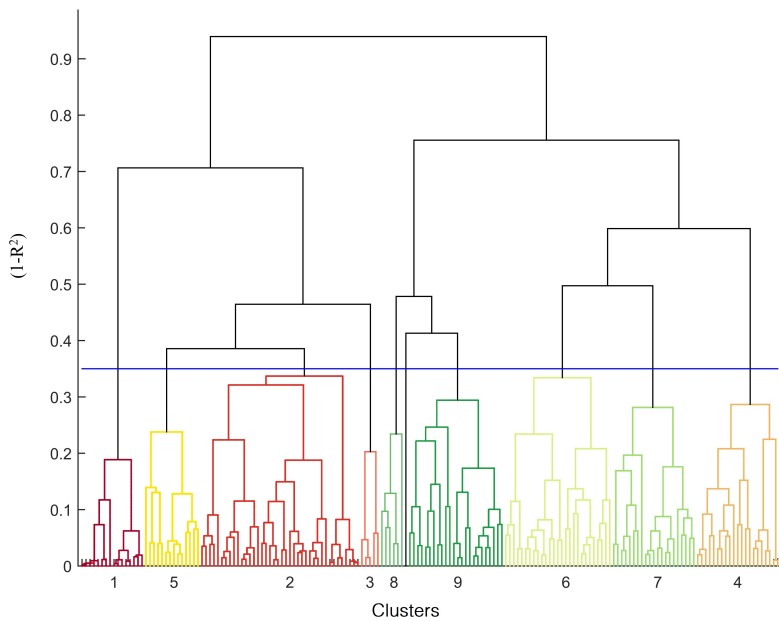

**Figure 6.** Dendrogram results: cluster colours are the same as in Fig. 7, for better identification. X axis numbers are the cluster's numbers. Y axis is $(1-R^2)$, where r is the correlation coefficient between FTBA and DSRp. The blue line is the clustering threshold, at 0.35. Each vertical line is a municipality.

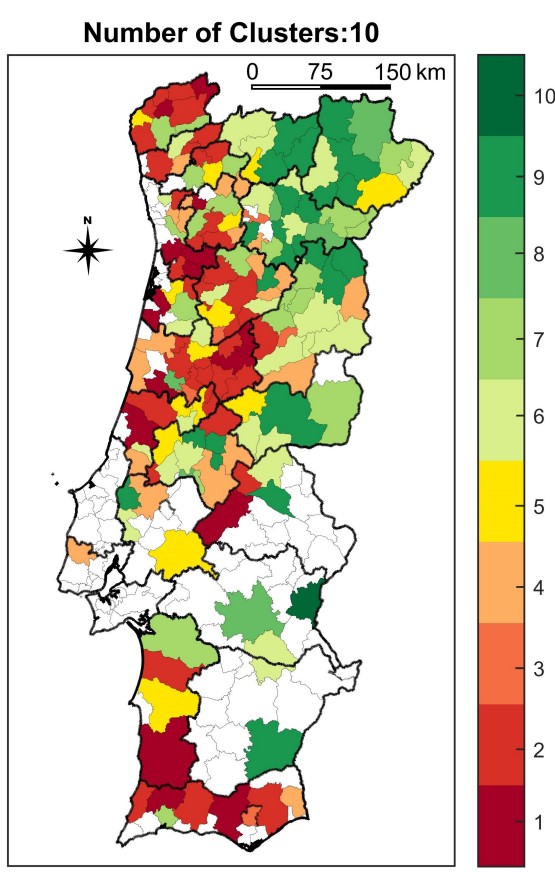

**Figure 7.** Clusters spatial distribution. Cluster colours are the same as in Fig.6. Municipalities without colour were excluded from the cluster analysis, justifying only 5.2% of TBA

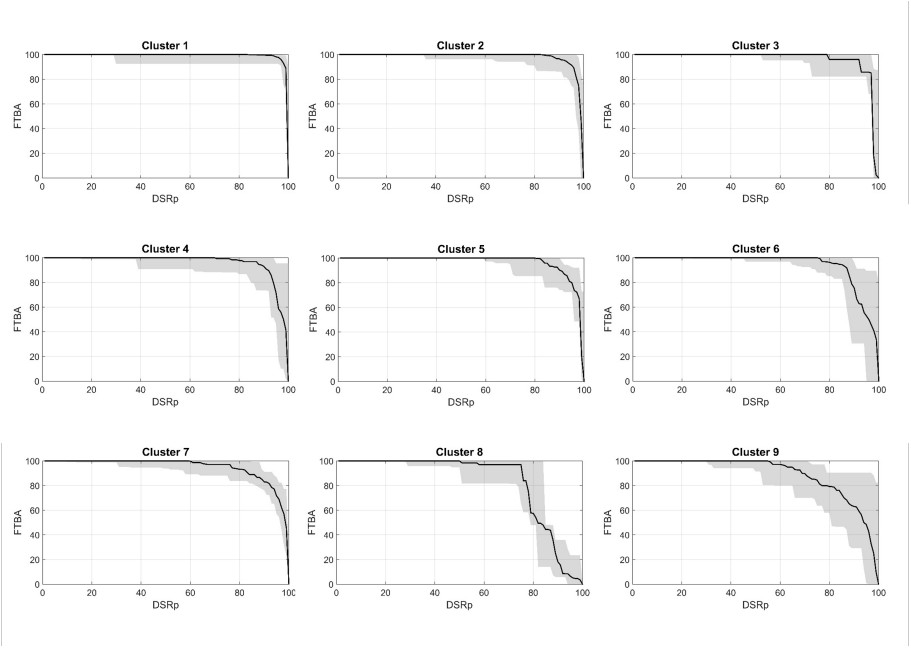

**Figure 8.** Fraction of total burnt area (FTBA) *vs.* DSR percentile (DSRp), for the municipalities of each of the 9 clusters. The black line is the median of all curves in each cluster. The shaded area is defined by the maximum and minimum curves in each cluster.

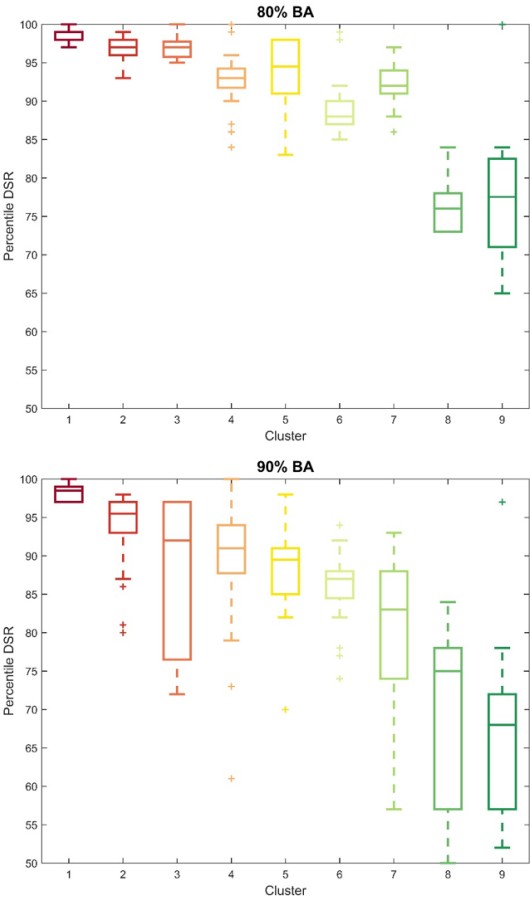

**Figure 9.** Boxplots of DSRp80TBA (top panel) and DSRp90TBA (bottom panel), i.e., the DSRp associated with 80% and 90% of TBA, respectively, for the 9 clusters. The central line is the median; the edges of the box are the 25th and 75th percentiles; and, the plus signs represent outliers, defined as a value that is more than three scaled median absolute deviations away from the median.

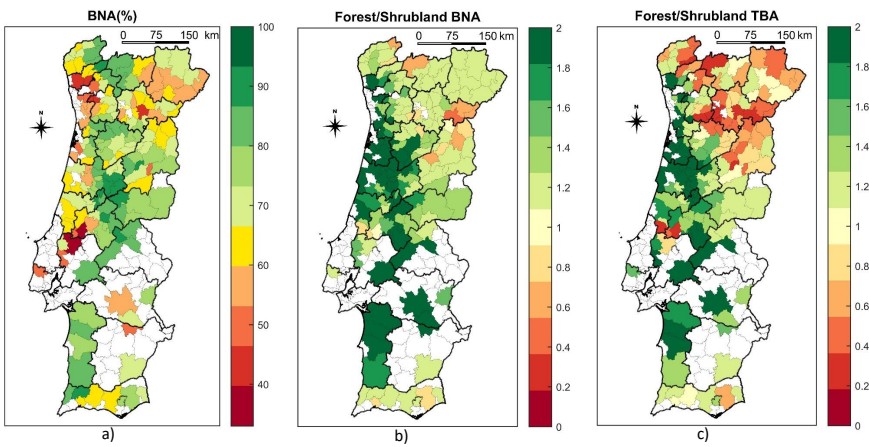

**Figure 10.** a) Burnable area (BNA), in percentage; b) Forest/Shrubland burnable area (BNAF/BNAS) and c) Forest/Shrubland total burnt area (TBAF/TBAS); all in the 2001 – 2019 period, for the selected municipalities.

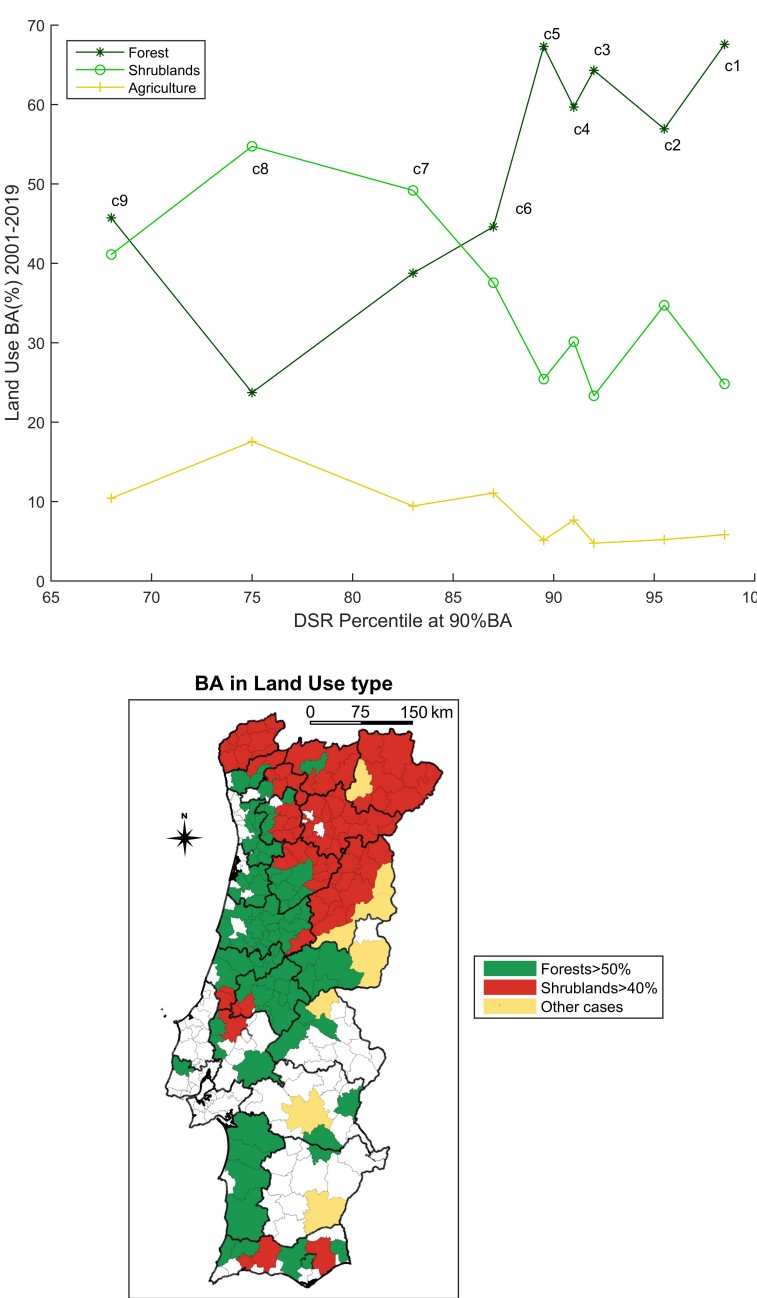

**Figure 11.** Top: Burnt area (BA) in three land use types: forest, shrublands and agriculture; represented for each cluster, identified by the respective DSRp and also by letter c. Bottom: Municipalities with Burnt Area in Forest>50%, Shrublands>40% or other cases. Municipalities without colour were excluded from the cluster analysis.

**Table 1.** Data sources, types, variables and methodology where it is used.

| Data source and type | Variables | Methodology |
|---|---|---|
| Wildfire data for 2001 – 2019. Provided by the *Instituto da Conservação da Natureza e das Florestas* | Burnt area (BA) polygons for wildfires with BA > 100 ha | To compute burnt area metrics (Table 2) |
| ERA5-Land. Meteorological data for 2001-2019. Provided by the ECMWF | Temperature<br>Relative Humidity<br>Wind Speed<br>Precipitation | To compute FWI indices, including DSR |
| COS2018. Land Use and Land Cover data. Provided by the *Direção Geral do Território* | Forest<br>Shrublands<br>Agriculture<br>Agroforestry<br>Other burnable areas | To assess burnable areas and the land cover type affected by each wildfire |

**Table 2.** Burnt area metrics used in the manuscript, including acronym, definition and spatial scale of application/use.

| Burnt area metric | Definition | Scale |
|---|---|---|
| Total Burnt Area (TBA) | $TBA = \sum_{i=1}^{n} BA_i$   $n$ =total number of wildfires | National and Municipal |
| Log(accumulatedBA) | $FTBA = 100 - (\frac{\sum_{i=1}^{m} BA_i}{TBA} \times 100\%)$ <br> $m$ =number of sampled wildfires | National |
| Fraction of Total Burnt Area (FTBA) | $FTBA = 100 - (\frac{\sum_{i=1}^{m} BA_i}{TBA} \times 100\%)$ <br> $m$ =number of sampled wildfires | National and Municipal |
| DSR percentile associated with 90% of TBA (DSRp90TBA) | $DSRp90TBA = DSRp(0.90 \times TBA)$ | National and Municipal |
| DSR percentile associated with 80% of TBA (DSRp80TBA) | $DSRp80TBA = DSRp(0.80 \times TBA)$ | National and Municipal |
| Burnable Area (BNA) | $BNA = \frac{\text{Area of burnable land cover type}}{\text{Total area}} \times 100\%$ | Municipal |
| BNAF/BNAS | $\frac{\text{Forest BNA}}{\text{Shrubland BNA}}$ | Municipal |
| TBAF/TBAS | $\frac{\text{Forest TBA}}{\text{Shrubland TBA}}$ | Municipal |
| Burnt Area in Forest (BAF) | $BAF = \sum_{i=1}^{f} BA_f$ in Forest areas <br> $f$ =number of wildfires that occurred in Forest | Cluster |
| Burnt Area in Shrubland (BAS) | $BAS = \sum_{i=1}^{f} BA_s$ in Shrubland areas <br> $f$ =number of wildfires that occurred in Shrubland | Cluster |
| Burnt Area in Agriculture (BAA) | $BAA = \sum_{i=1}^{f} BA_a$ in Agricultural areas <br> $f$ =number of wildfires that occurred in Agriculture | Cluster |

**Table 3.** Contingency tables and accuracy metrics to assess the role of vegetation Burnt Area (BA) assessed with DSRp90BA thresholds, for the municipalities used in cluster analysis. The contingency tables computed the number of municipalities (NM) for the following criteria: CLUST 1-5 (CLUST 7-9) and BAF>50% (BAS+BAA>50%). Overall Accuracy (OA), User's Accuracy (UA) and Producer's Accuracy (PA) were the calculated accuracy metrics, together with the statistical tests Chi-squared ($\chi2$) test (with p-value), Phi coefficient ($\Phi$), Contingency coefficient (C) and the Cohen's Kappa coefficient ($\kappa$).

| NM | BAF>50% | BAS+BAA>50% |
|---|---|---|
| CLUSTERS 1-5 | 65 | 27 |
| CLUSTERS 7-9 | 14 | 33 |
| OA | 71% | |
| UA | 71% | 70% |
| PA | 82% | 55% |
| $\chi2$ | 21.175 (4E-6) | |
| $\Phi$ | 0.390 | |
| C | 0.363 | |
| $\kappa$ | 0.383 | |

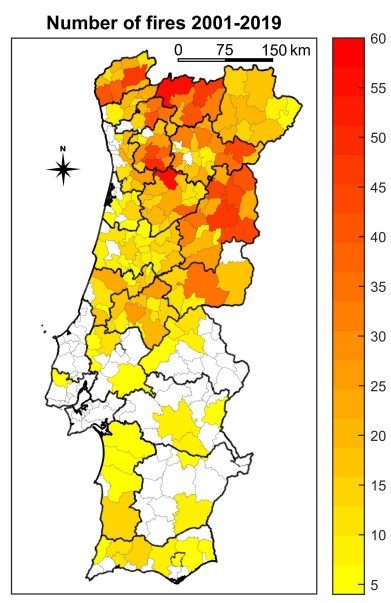

**Figure A1.** Number of fires larger than 100 ha, all in the 2001 – 2019 period, for the selected municipalities.