# Peer review of "Drivers of extreme burnt area in Portugal: fire weather and vegetation"

_Natural Hazards and Earth System Sciences, 2021_

## Author Response (AR1)

**RC1: 'Comment on nhess-2021-173', Anonymous Referee #1, 04 Aug 2021**

Citation: https://doi.org/10.5194/nhess-2021-173-RC1

**Anonymous Referee #1:** The paper "Spatial variability in the relation between fire weather and burned area: patterns and drivers in Portugal" aims at analysing large (>100 ha) summer forest fires in mainland Portugal. Although the dataset is sound and of potential interest, I found several flaws in the methods and results interpretations. The use of statistics is not always appropriate, expecially the use of OA, UA and K (they are used for classification accuracy against a reference.

**Answer:** We want to start by thanking the reviewer; first, for the work and time spent on reviewing our manuscript, as we recognize the difficulties that the editors have in finding good reviewers; second because your comments and suggestions were very helpful in clarifying and strengthening the manuscript.

The reviewer states that the objective of this work was to "analyze large (>100 ha) summer forest fires in mainland Portugal". We comprehend that the objectives were not clear and clarified the three objectives of this study in lines 2-4 (Abstract) and lines 95-99 (Introduction) were to 1) assess if the DSR90p threshold is adequate to identify the bulk of burned area (BA) for mainland Portugal; 2) identify and characterize regional variations of the DSRp threshold that justify the bulk of BA, and; 3) analyze if vegetation cover can explain the spatial variability of the DSRp threshold.

We do not agree with the reviewer when he states that the use of OA, UA and K were not appropriate. Effectively, these statistics were used for classification accuracy against a reference as the reviewer proposes. This was explained in the section 2.4, together with a detailed explanation of the accuracy metrics and the citations of previous articles that defined and used the same statistics in similar conditions/situations. However, we agree that this reference may be insufficient and provide a better explanation. Therefore, we changed the manuscript accordingly, in lines 216-232. Additionally, we improved the explanation and justification of these results, including more references, in lines 361-367.

We appreciate the reviewer's acknowledgement of the quality of the datasets. On one hand, the wildfire and land use data used in this manuscript are Portuguese official datasets, produced and provided by Portuguese national authorities. These datasets were used in many other studies, by a large number of authors for a wide variety of purposes (Bergonse *et al*., 2021, Tarín-Carrasco *et al*., 2021). On the other hand, the ERA5 is recognized as the best or one of the best global atmospheric reanalysis datasets (Huai et al., 2021, Muñoz-Sabater *et al*., 2021, Urban *et al*., 2021) and used worldwide (Chinita *et al*, 2021, Sianturi *et al*., 2020). Therefore, it is one of the most used meteorological datasets in the world. We added this information in the manuscript, in lines 121-127.

Citations included in the manuscript:

Bergonse, R., Oliveira, S., Gonçalves, A., Nunes, S., DaCamara, C. and Zêzere, J.L.: (2021) Predicting burnt areas during the summer season in Portugal by combining wildfire susceptibility and spring meteorological conditions, Geomatics, Natural Hazards and Risk, 12:1, 1039-1057, DOI: 10.1080/19475705.2021.1909664, 2021.

Chinita, M.J., Richardson, M., Teixeira, J. and Miranda, P.M.A.: Global mean frequency increases of daily and sub-daily heavy precipitation in ERA5. Environ. Res. Lett., 16, 074045, https://doi.org/10.1088/1748-9326/ac0caa, 2021.

Huai, B., Wang, J., Sun, W., Wang, Y. and Zhang, W.: Evaluation of the near-surface climate of the recent global atmospheric reanalysis for Qilian Mountains, Qinghai-Tibet Plateau, Atmospheric Research, 250, https://doi.org/10.1016/j.atmosres.2020.105401, 2021.

Muñoz-Sabater, J., Dutra, E., Agustí-Panareda, A., Albergel, C., Arduini, G., Balsamo, G., Boussetta, S., Choulga, M., Harrigan, S., Hersbach, H., Martens, B., Miralles, D. G., Piles, M., Rodríguez-Fernández, N. J., Zsoter, E., Buontempo, C., and Thépaut, J.-N.: ERA5-Land: a state-of-the-art global reanalysis dataset for land applications, Earth Syst. Sci. Data, 13, 4349–4383, https://doi.org/10.5194/essd-13-4349-2021, 2021.

Sianturi, Y., Marjuki and Sartika, K.: Evaluation of ERA5 and MERRA2 reanalyses to estimate solar irradiance using ground observations over Indonesia region. AIP Conference Proceedings 2223, 020002, https://doi.org/10.1063/5.0000854, 2020.

Tarín-Carrasco, P., Augusto, S., Palacios-Peña, L., Ratola, N. and Jiménez-Guerrero P.: Impact of large wildfires on PM10 levels and human mortality in Portugal. Nat. Hazards Earth Syst. Sci., 21, 2867–2880, https://doi.org/10.5194/nhess-21-2867-2021, 2021.

Urban, A., Di Napoli, C., Cloke, H. L., Kyselý, J., Pappenberger, F., Sera, F., Schneider, R., Vicedo-Cabrera, A.M., Acquaotta, F., Ragettli, M.S., Íñiguez, C., Tobias, A., Indermitte, E., Orru, H., Jaakkola, J.J.K., Niilo, R.I.R., Pascal, M., Huber, V., Schneider, A., de' Donato, F., Michelozzi, P., Gasparrini, A.: Evaluation of the ERA5 reanalysis-based Universal Thermal Climate Index on mortality data in Europe, Environmental Research, Volume 198, 111227, ISSN 0013-9351, https://doi.org/10.1016/j.envres.2021.111227, 2021.

**Anonymous Referee #1:** Furthermore no analyses on the spatial pattern are provided, simply a description of the maps. The relationships between LULC and weather and fire occurence need to be deepen analysed. The authors need to clarify and explain better the reason why they divided in cluster the dataset, the significant differences between groups,.... Other multivariate analysis methods can be adopted considering more variables (e.g. geographical gradients, inhabitants,...) in order to obtain a strong explanatory analysis and then building relationships or models to be tested.

**Answer:** We do not agree with the first phrase in this comment of the reviewer. First, we have some difficulty understanding it, because the reviewer does not specify what spatial pattern he refers to. Second, any spatial pattern that the reviewer may refer to are (the maps of) the results obtained with the selected methods that, obviously have to be described and discussed (in the adequate sections), but they do not necessarily have to be subject to additional analysis with another methodology, because they are obvious and easy to interpret. Third, the spatial pattern of the DSRp was deeply analyzed using cluster analysis, with an explanation of the dendrogram based on the correlation coefficient between FTBA and DSRp in each of the municipalities (Figure 7); and, additionally, showing and explaining the differences of the curve FTBA vs DSRp, in each cluster (Figure 9). We also validated the relationship between the LULC and the DSRp (and, consequently, with the clusters) using the contingency tables and statistical tests. We consider that this analysis was sufficient to explain and justify the results of this manuscript. However, we understand that the explanation was not sufficient and we will present additional justifications in the following lines and also in the manuscript.

It is important to underline that this study is not about the relationship between LULC and weather and fire occurrence. In summary, this study is about the relationship between extreme fire weather and high/extreme burnt area which slightly change spatially due to LULC. The relationship between fire weather and the fire incidence has been analyzed by many researchers for several years, as mentioned in the manuscript (please, see lines 17-27, 380-385). The role of LULC in the incidence of fire has also been analyzed and some references on this subject are cited in the manuscript. The authors of this manuscript are also authors of articles published on these relationships, but many more references could be provided. However, in this study, we intended to carry out a deeper analysis between fire weather and the incidence of fire and ended up discovering that this relationship depends on LULC, but does not depend on any of the other more important factors of the fire incidence, as described in lines 246-259, and also motivated us for the analysis described in sections 3.2, 3.3 and 3.4.

It should be noted that, while LULC, topography, population statistics, etc. are structural (essentially fixed or stationary) wildfire hazard factors, the meteorological conditions are conjunctural (essentially variable or dynamic) wildfire hazard factors. Despite a few space-time analyses (e.g., Orozco et al., 2012; Pereira et al. 2015, Parente et al., 2016), usually, and for obvious reasons, the influence of these two types of factors on the fire incidence is studied separately.

However, it was precisely as a result of an in-depth analysis of the relationship between extreme fire weather (specifically DSRp) and fire incidence (specifically the burnt area) that it was possible to conclude that LULC - a structural factor - influences the impacts of meteorological conditions - a conjunctural factor of fire risk. As far as we know, this is the first study that identifies and establishes that the relationship between fire weather and fire incidence depends on LULC, for Portugal.

It is also important to underline that, to establish this relationship, we used objective methods and adequate statistics that ensure the robustness and statistical significance of the results. The description of the study carried out also includes the chronology of the performed analysis. In a previous study/paper (Calheiros et al., 2020), the relationship between fire weather and fire incidence was analyzed in-depth for the entire Iberian Peninsula. Among other results, we found that the DSR90p is a good indicator of extreme fire weather and well related to the burnt area in the Iberian Peninsula, as mentioned in lines 47-49. In this study, we started by verifying whether the relationship between DSRp and BA found, in general terms, for the Iberian Peninsula, was also verified in Mainland Portugal, at municipality level, and what is the spatial variability of the extreme value of DSRp above which most of the burned area is registered (objective 1 and 2 of this manuscript). To assess the spatial variability, we based our analysis on the 278 municipalities of mainland Portugal. To objectively interpret the obtained spatial patterns (Figure 5), we complemented and deepened the analysis with the use of clustering algorithms, to classify the municipalities into statistically different groups in terms of the relationship between FTBA and DSRp. Based on our knowledge and experience, we tested all the most likely factors that could help to explain the obtained results. The emerging patterns showed that all of those most likely factors, such as topography, altitude (Figure 1); slope (please see Figure 5 of Parente and Pereira (2016)); population density (please see Figure 2 of Pereira et al.(2011) or Figure 2 of Parente and Pereira (2016)); rural and urban area type (please see Figure 3 of Pereira et al. (2011)); road density/distance to the nearest road (please see Figure 2a of Parente et al.(2018)) and climate type (please see Figure 1a of Parente et al.(2016)) were not able to explain the obtained spatial patterns. The only factor with a similar spatial pattern was the LULC, which is the reason why we decide to explore this possibility more deeply, with contingency tables and several accuracy metrics to assess the influence of the type of vegetation cover

on the relationship between DSRp and TBA (as described in lines 209-232). Therefore, we agree that the explanation was not clear in the text, so we changed the manuscript accordingly, regarding this discussion/subject in lines 430-456 (Discussion section).

One of the objectives of this manuscript was to identify and characterize regional variations of the DSRp threshold that justifies the bulk of BA. The analysis was performed on the spatial basis of the municipalities. The high number (278) of these administrative regions difficults the interpretation of the results and the objective and statistically significant assessment of differences between the results obtained for different municipalities. The cluster analysis was performed to execute and simplify this task, but also to identify the major macro scale patterns. We agree with the reviewer that this procedure was not clear and added a phrase to clarify and explain better the reason why we decide to use cluster analysis in subchapter 2.3, in lines 179-181. We want to underline that the significant differences between groups were clarified and explained along with the dendrogram, in Figure 7, the spatial distribution of the clusters, in Figure 8, as well as in the text, especially in subchapter 3.3.

Furthermore, we improved the explanation of the dendrogram and clustering procedure as follows:

The following notation describes the linkages (the distance between two clusters) used in the *complete* clustering method (MathWorks, Inc.):

- Cluster $r$ is formed from clusters $p$ and $q$.

- $n_r$ is the number of objects in cluster $r$.

- $x_{ri}$ is the $i$th object in cluster $r$.

- *Complete linkage*, also called the *farthest neighbour*, uses the largest distance between objects in the two clusters (Eq.1).

$$d(r,s) = max\left(dist\left(x_{ri}, x_{sj}\right)\right), i \in (1, \dots, n_r), j \in (1, \dots, n_s) \tag{1}$$

A distance metric is a function that defines a distance between two observations. The Matlab function *pdist* used in this study, that computes the pairwise distance between pairs of observations, supports various distance metrics, namely: Euclidean distance, standardized Euclidean distance, Mahalanobis distance, city block distance, Minkowski distance, Chebychev distance, cosine distance, correlation distance, Hamming distance, Jaccard distance, and Spearman distance. We used the correlation distance in this article:

Given an $m$-by-$n$ data matrix X, which is treated as $m$ (1-by-$n$) row vectors $x_1$, $x_2$, ..., $x_m$, the correlation distance between the vector $x_s$ and $x_t$ are defined as in Eq.2:

$$d_{st} = 1 - \frac{(x_s - \overline{x_s})(x_t - \overline{x_t})'}{\sqrt{(x_s - \overline{x_s})(x_s - \overline{x_s})'}\sqrt{(x_t - \overline{x_t})(x_t - \overline{x_t})'}}, \tag{2}$$

where $\overline{x_s}$ is described in Eq.3:

$$\overline{x_s} = \frac{1}{n}\sum_j x_{sj} \ \ and \ \ \overline{x_t} = \frac{1}{n}\sum_j x_{tj}. \tag{3}$$

The selected (1-r2) threshold was 0.35, meaning that the coefficient of determination in the municipalities within the same cluster is higher than 0.65. This value was selected after the analysis of the dendrogram and results from the balance between the correlation between municipalities and the total number of clusters. For example, on one hand, if we have chosen 5 clusters, the correspondent correlation between municipalities within the same cluster will be larger than 0.5, a value that we considered too low for this analysis. On the other hand, for a higher correlation, for example, 0.75, which corresponds to 1-r2=0.25, the number of clusters will be much higher, increasing the difficulty of interpreting the maps and dendrogram. The explanation of the dendrogram and clustering procedure is provided in the new version of the manuscript (in lines 182-204).

Finally, as appointed in manuscript (lines 41-44), we want to highling that: "Cluster analysis for the Iberian Peninsula has identified several regions with similar fire regimes, using several variables related to fire, as intra-annual pattern of burnt area (Trigo et al., 2016; Calheiros et al., 2020; Calheiros et al., 2021), fire activity and weather risk (Jimenez-Ruano et al., 2018), large fire-weather typologies (Rodrigues et al., 2020) or burnt area tendency (Silva et al., 2019)".

Citations included in a new version of the manuscript:

Orozco, C. V., Tonini, M., Conedera, M., & Kanveski, M. (2012). Cluster recognition in spatial-temporal sequences: the case of forest fires. *Geoinformatica*, 16(4), 653-673.

(The other citations were already listed in the manuscript)

**Anonymous Referee #1:** The discussion section is mainly a list of municipalities in the different groups or sentences related to parameters not considered in the study. The first part is belonging to methods.

**Answer:** We believe that the previous version of "Discussion" fulfils what is expected in this section of an article, namely justifying, validating and interpreting options (data and methodology) and results, based on the findings of previous studies. In the submitted version of the manuscript, we started by discussing the methodological options (that we moved to the Methods, as suggested by the reviewer, now in lines 157-166). Then we discussed the characteristics (including the limitations) of the LULC dataset and the potential impacts on our study (now in Methods, lines 167-176). Then, we discussed the obtained results, presenting the justification, interpretation and validation of the findings, in line with previous studies (now in lines 380-429). As the study was carried out using municipalities as a spatial unit, we assumed that it is expected that there will be references to some municipalities. However, these references to the municipalities only occur in 4 of the previous 68 lines of the discussion, which represents less than 6% of the total number of lines of the discussion. Nevertheless, we made changes to the discussion section (also deleted the previous references to municipalities) and add paragraphs regarding other issues appointed by the reviewer. The new paragraphs of the discussion are in lines 430-456.

We agree with the reviewer that the first part of the discussion belongs to the Methods section and, therefore, we changed the manuscript accordingly, in lines 157-176.

**Anonymous Referee #1:** The conclusion is redundant, repeating results and discussion elements.

**Answer:** We agree that the text has repeated ideas, results and conclusions. Consequently, we changed this section according to the suggestion of the reviewer, now in lines 458-479.

**RC2: 'Comment on nhess-2021-173', Anonymous Referee #2, 28 Oct 2021**

Citation: https://doi.org/10.5194/nhess-2021-173-RC2

**Anonymous Referee #2:** The manuscript from Calheiros et al is very difficult to read. It reads as a series of bullet points linked together and the authors did not even bother at breaking the text into paragraphs or making any effort to increase readability. This is not a result of the authors not being native English speakers (I believe paragraphs also exist in Portuguese) but rather denotes a major lack of attention to detail.

**Answer:** First of all, we want to thank the reviewer for his pertinent commentaries that contribute to improving this manuscript. We deeply regret the absence of paragraphs. The manuscript was written with paragraph breaks. We suspect that paragraphs were deleted by the software used for the submission of the manuscript. We corrected it in the new version of the manuscript.

Regarding this subject, we also changed the Introduction. It is important to refer that this modification did not change the subject matter, only changed the way of writing. The Introduction is now in lines 14-99.

**Anonymous Referee #2:** The authors examine thresholds in burnt area associated with DSRp and how they differ across Portugal. The way they present the data is somewhat misleading: they make us believe that DSR has a very high correlation with burnt area. These types of correlations have been described before and they result from the ordering of values (from small to large). If that order is removed and simple scatter plot of burnt area DSR is presented, that relationship usually breaks, or is much weaker. I would thus encourage the authors to be more careful when using these types of analyses.

**Answer:** We disagree with the reviewer but we believe that the manuscript does not explain clearly this subject, so we change it accordingly. In particular, we agree that the DSR vs BA scatter plot does not reveal a simple robust relationship between these two variables. Please see the figure below (also added to the manuscript – Figure 2), where the logarithm of the burnt areas - Log(BA) - is plotted against the percentiles of DSR. This is due to several reasons (e.g., ignition source, firefighting activities, geographical/landscape features, fire barriers, limitations of the Fire Weather Index System to represent the role of fire weather drivers, humidity of live fuel moisture and the convective influence in fire behaviour, etc.) but, in essence, the most important one is that the wildfire activity does not only depend on the weather. This means that: (i) wildfires can occur in days with relatively low values of DSR; (ii) small wildfires can occur in days of high DSR, due to rapid fire-suppression activities or other constraints (especially fuel). However, it is well known that extreme wildfires only occur in days of extreme fire weather (Fernandes et al., 2016). These facts are validated by our results, revealing that only 6% of the Total Burnt Area (TBA) occurs with DSRp<80 and 12% of TBA is registered in wildfires with DSRp<90. These reasons explain all the main features of the figure below, namely: small wildfires are registered in days with almost all values of DSR, although the much small number of wildfires in the lower left quarter of the plot area, and the huge number of events near the right vertical axis, especially for DSR>DSR90p. In effect, DSR seems to act as an upper limit to the maximum burnt area. It is precisely the relationship between the burnt area and the DSR in this "region"

of the plot that is investigated in this study. This is clearly explained in the manuscript and illustrated in Figure 3 and Figure 4. We added this clarification in the manuscript, in lines 369-379.

Furthermore, we'd like to add that cumulative statistics are commonly used, including in wildfire science. See, for example, Cumming, S. G. (2001). A parametric model of the fire-size distribution. Canadian Journal of Forest Research, 31(8), 1297-1303.

Jiang, Y., & Zhuang, Q. (2011). Extreme value analysis of wildfires in Canadian boreal forest ecosystems. Canadian journal of forest research, 41(9), 1836-1851.

Kanevski, M., & Pereira, M. G. (2017). Local fractality: The case of forest fires in Portugal. Physica A: Statistical Mechanics and its Applications, 479, 400-410.

[Figure]

**Anonymous Referee #2:** I'm not familiar with the clustering techniques used by the authors, and I will not comment on those. I will just point out that the results are rather shocking because pretty much all clusters are distributed across all Portugal, but it is well known that fires in N PT differ substantially from S PT (the authors actually state this in their introduction as well).

**Answer:** A better explanation of the purpose of cluster analysis and the aim of its application in this study can help understand the results.

We want to clarify the three objectives of this study (as stated in lines 2-4 of the Abstract and lines 95-99 in the Introduction) were to "1) assess if the DSR90p threshold is adequate to identify the bulk of burned area (BA) for mainland Portugal; 2) identify and characterize regional variations of the DSRp threshold that justify the bulk of BA, and; 3) analyze if vegetation cover can explain the spatial variability of the DSRp threshold ". If we had performed a cluster analysis on the number of fires or burnt areas, the results would be clusters in regions where the incidence of fire is higher, ie in the central-north and extreme south region (Algarve) as the reviewer suggests. Results of this type of study can be consulted, for example, in:

Pereira, M. G., Caramelo, L., Orozco, C. V., Costa, R., & Tonini, M. (2015). Space-time clustering analysis performance of an aggregated dataset: The case of wildfires in Portugal. Environmental Modelling & Software, 72, 239-249.

Parente, J., Pereira, M. G., & Tonini, M. (2016). Space-time clustering analysis of wildfires: The influence of dataset characteristics, fire prevention policy decisions, weather and climate. Science of the total environment, 559, 151-165.

Kanevski, M., & Pereira, M. G. (2017). Local fractality: The case of forest fires in Portugal. Physica A: Statistical Mechanics and its Applications, 479, 400-410.

Some of these papers are already cited in the manuscript.

However, as explained in the manuscript in lines 6-7 and now in lines 178-181, the cluster analysis was motivated by the "spatial distribution of DSRp80TBA and DSRp90TBA" (lines 258-259), using a methodology described in section 2.3 (now in lines 179-205), "based on the DSRp vs FTBA curves aimed to find groups of municipalities with similar fire-weather relation" (lines 412-413) i.e., to group the municipalities that present similar relationship between DSRp and TBA and help to explain "some important differences appear among DSRp thresholds that explain 90 and 80% of the TBA" (lines 460-461). The results were extensively described in section 3.3 (lines 290-328) which allow us to easily understand the purpose of having performed this analysis. In summary, the cluster analysis "revealed that municipalities where large wildfires occur in high DSRp present higher BA in forests and are located in coastal areas. In contrast, clusters with lower DSRp present greater BA in shrublands and are situated in eastern regions." (lines 10-12).

Regarding the pertinent reviewer comment, we improved the explanation of the cluster analysis, in lines 182-204.

---

## Referee Report (RR1)

[revised manuscript text omitted]

***Summary:***

Overall, this manuscript presents sound scientific ability and strong technical analytical skills, however, I have some significant doubts about the contribution to new science and the bridging of knowledge gaps to the field. Not only are the differences between this study and previous research not strongly argued but the lack of further contextual information about fire behaviour is an issue. The largest is the lack of consideration of local scale conditions on relatively small-scale fires ~100ha. This study uses regional scale weather inputs to assess potentially local scale drivers and influences. There is also a lack of understanding fuel characteristics that drive fire behaviour and their relationship to occurrence.

The authors make an effort to address purpose of this research (through its relevance to Portugal) but it fails to outline the significance of its work in relation to other similar fire prone regions internationally and there is a lack of reference material to the broader fire science community, to which have produced similar work (conceptually).

There are many missing definitions and structural issues with this manuscript, I'd strongly suggest that the authors thoroughly revise the manuscript to improve the communication of the aims and objectives. The conclusions are also not strongly supported nor adequately explained. I'd also suggest that there needs to be some further work to highlight the novel nature of the work, particularly when LULC and fire occurrence is so broadly researched internationally.

***Major Comments:***

Line 40: The DSR is actually never properly addressed in terms of its composition and its strength/weaknesses versus other fire weather indices. There is a lack of depth when considering other potential metrics as well as almost no discussion about how these are produced or applied.

Line 60: The understanding of vegetation and its role in driving large scale fires is poorly discussed. There is no reference to grasslands or ephemeral grasses, as well as many other prominent vegetation types. Further to this, I'd be certain that on the 100ha scale that Eucalypts behave very differently than other native vegetation. It is also mentioned that Eucalypts are not a significant driver in the change in fire regime, however it makes up the largest percentage of fuels in Portugal. This needs to be explained better.

Line 90: I'm not convinced about the gaps in knowledge. I don't think there has been enough effort to explain the differences between what is proposed and previous studies.

Line 95: There has been almost no effort to discuss vegetation cover and what metric will be used to assess it.

Line 135: Is 1200UTC the most suitable reference time for Portuguese fires? Peak conditions are typically around 1400 local time?  This has not been discussed at all.

Line 140:  9km is big when considering 100ha fires. Your weather data is roughly 9 times more course than the ignition data? I know this study is generally considering regional drivers, however, this is not robustly considered or discussed. Perhaps but is probably not a good dataset for considering regional scale fire weather and ignitions. Also, you have listed just two "worldwide" studies.

---

## Author Response (AR2)

Report #1

Submitted on 25 Mar 2022

**Anonymous referee #3**

This study focuses on assessing the Daily Severity Rating (DSR), an additional component of the Canadian Fire Weather Index (FWI), to provide insight into its suitability for summer large fires in Portugal and the role of vegetation. The manuscript states that filling this knowledge gap can support the study region's fire prevention and suppression planning. Overall, the paper was informative. However, there were several issues with clarity, conciseness and structure, which makes it hard to evaluate the validity of the methods, results and the overall study.

**Answer:** We want to start by thanking the reviewer for the work and time spent on reviewing. The reviewer's constructive comments and suggestions were very helpful in clarifying and strengthening the manuscript, that we believe is now improved the quality and, therefore, the suitability for publication.

**Anonymous referee #3:** Some major general points:

1. The title would spark more interest keeping it shorter and removing redundancy. The authors should consider editing the title to be concise and relevant.

**Answer:** We edited the title, making it more concise as the reviewer suggested.

2. The abstract explained the aims and tasks of the research and the main findings. However, authors should improve the context and state the research's need and conclusion to make it easy to understand to less specialised audiences.

**Answer:** The abstract was changed, in particular the conclusion and introduction, taking into account the limit of word's number.

3. The introduction provided enough background and a clear knowledge gap that the study aims to address. However, to improve the understanding, the authors should reorganise the background topics into a logical, simple, and clear structure, avoiding talking about the same point multiple times throughout the introduction. For example, the authors could move from the Mediterranean context to Portugal, join fire weather and indices background, and combine vegetation with land uses changes, referring briefly to the urban-rural interface as this driver is not the focus of the study. The last paragraph of the introduction could mention the study of Calheiros et al. (2020), indicate the knowledge gap, what the authors did to fill the gap, and finally, an overview of the paper's contributions.

**Answer:** We agree with the reviewer that the Introduction was not well structured, so we change it accordingly. We highlight that, in the track change file, it shows that most of introduction is new. However, that is not true. We generally reorganized the Introduction, now in lines 14-94.

4. The authors could complete the study area description providing information about the vegetation across Portugal. Additionally, as the map of figure 1 shows pyroregions, the authors should refer to them in the text or remove them. Finally, the last sentence, lines 108-110, should be moved to the introduction (e.g., before mentioning the findings from Calheiros et al. (2020) in the introduction) or discussion.

**Answer:** Two phrases regarding the vegetation in Portugal were added, in lines 103-106. Additionally, we added a brief description of the pyro-region, in lines 107-109. We agree with the reviewer that the last sentence should be moved. However, we changed the sentence to better include in the new Introduction's structure. This subject is now better explained in lines 95-103.

5. The methods section should be reorganise to present the data and the approaches in a clear structure. For example, the author could use tables to list the data sources and the burned areas metrics. Furthermore, split the information into focused and clear subsections. A similar structure should be followed when reporting the results. The discussion should focus on the main findings.

**Answer:** We consider that this reviewer's commentaries were very constructive and useful for this manuscript's improvement.

Firstly, we created two tables to list data sources and to clarify burned areas metrics. In particular, Table 1 describes the data sources, types, variables and methodology where it is used. Table 2 describes the burnt area metrics used, including acronym, definition and spatial scale of application or use. Both tables were included in pdf file format (in page 33 and 34).

Secondly, we slip the information into more focused and clear subsections. In particular, we created new Data and Methodology subchapters, and the Results are also shown in the same subsections. The Data and Methodology chapter is now between lines 95 and 219. The Results chapter is now in lines 220-345.

Thirdly, we changed the Discussion as the reviewer suggested. We believe that the Discussion in now more focused on the main findings. The Discussion is now in lines 346-417.

6. The conclusion should be more general and concise. The authors should avoid repeating the results and focus on the meaning of the results and perhaps implications and perspectives. As a first step, they could focus only on the last paragraph and remove the earlier information.

**Answer:** We changed the conclusion as the reviewer suggested. The conclusion is now more general and concise, in lines 419-430.

7. To improve the quality of the figures, the authors should be consistent using the same map layout, the terms (e.g., r2 vs R2) and plot frames. Furthermore, they could make all the plots bigger, remove titles on the top of the plots and ensure the caption descriptions.

**Answer:** We changed the figures as the reviewer's suggested. In particular, we changed the Figure 2, because we believe that the new format can explain our work in a much better way than the previous one. We increased the file original size and resolution in all Figures. However, we want to highlight that the Figures in pdf are all sized by the LaTeX template to create the pdf.

8. The references are relevant, and appropriate key studies are included. However, authors should review and correct the list. For example, on page 18, line 569:679, remove the link.

**Answer:** We corrected the references, taking into account the reviewer's suggestion.

9. The English language is simple. However, some odd constructs that, if rephrased, would improve the quality and flow of the manuscript's text and make it easy to read. The manuscript should get proofread in English.

**Answer:** We carefully reviewed the manuscript and corrected several phrases.

**Anonymous referee #3:** Some minor points:

1. On page 2, line 37, to remove statistical concept.

**Answer:** The phrase was corrected and also moved to the beginning of Introduction, now in line 15.

2. On page 2, line 23, to add, e.g., in the references.

**Answer:** As previously, the phrase was corrected and moved, now in line 37.

3. On page 3, line 57:59, to provide references.

**Answer:** We modified the sentence to clarify that the references were provided before. The sentence is now in lines 60-61.

4. On page 3, line 88, to edit, for Portugal. Entire and Continental Portugal are redundant.

**Answer:** We agree with the reviewer that say in the same phrase "Entire" and "Continental" is redundant. However, we deleted part of the sentence and merged it with other. The new phrase regarding this subject is now in lines 82-84.

5. On page 3, line 89, change reveal to revealed.

**Answer:** As previously, we merged the two sentences and deleted some parts, including thw word "revealed". The new phrase is currently in lines 82-84.

6. On page 4, line 112:114, to re-edit.

**Answer:** We agree that this sentence was not clear and re-edited. The new phrase is in lines 111-112.

7. Define the first time using acronyms in the text and all the figure captions. For example, on page 4, line 117, to define ECMWF.

**Answer:** We regret that lack of attention of us. The ECMFW acronym is now defined in lines 115-116. Additionally, we reviewed all the acronyms in text and figure captions.

8. On page 5, line 122:123, remove the last sentence.

**Answer:** The phrase was removed as the reviewer suggested. Now this paragraph ends in line 121.

9. On page 6, line 167:176, to move to the discussion.

**Answer:** We agree with the reviewer that this paragraph should move to the discussion, now in lines 392-401.

10. On page 7, line 205, remove because the software was mentioned before.

**Answer:** The phrase was removed. The last paragraph ends in line 194.

11. On page 14, line 430:436, to re-edit, remove or move to the introduction, where the authors should be sure the study goal is clear.

**Answer:** We accepted the reviewer's suggestion and removed the paragraph.

---

## Editor Decision (ED2)

Review for *"Drivers of extreme burnt area in Portugal: fire weather and vegetation"* of T. Calheiros *et al.*

In this paper,  the relation between Daily Severity Rating percentile (DSRp) and  the total burned area (BA) in Portugal is studied, with the aim of understanding its smaller scale (municipal scale) behaviour.  The  Authors tried to  1) assess if the performance of  90th DSRp (DSR90p) threshold in BA prediction  in mainland Portugal; 2) identify and characterise regional variations of the DSRp threshold that justifies the majority of Burned Area ; and  3) analyse if the DSRp spatial variability could be explained with broad classification of land cover (forested vs agricultural vs shrubbed).

As a dataset, weather reanalysis data from ERA5-Land as well as wildfire and land use data from official Portuguese authorities for an extended summer period (15th May to 31st October) from 2001 to 2019 were used.

The treated topic is exceptionally relevant, since fire weather indices can and should be coupled by info on vegetation for optimal wildfire management procedures.

However, the paper should be refined in some parts before being considered for publication.

- Line 43 Define DSR (or at least specify that is a simple reformulation of FWI).  This can be done here or at line 135.
- Line 43 The reader needs to understand what a DSR percentile is. In order to get a threshold based on percentiles, we need a set of elements to be sorted in ascending order. On which set were the percentile classes defined?  This needs clarification.
- Line 115: how the burnt area dataset is derived? Polygons retrieved from ground assessments? Satellite?
- Line 115: of course, the threshold of 100 ha applies to European fire regime and not to, e.g., North American one.. Maybe this thought can be added in the text.
- Line 130 Why is air temperature and not air humidity the driver for fires?
- Line 145 How was the original classification of COS2018? How was the aggregation performed?
- Line 157 : "was allocated to **this** administrative unit". What unit are they talking about? The sentence can be reformulated.
- Line 160: Still not clear what BA percentages is.
- Line 163: So for each fire event at municipal scale, the maximum DSR is selected in the days of the event and the whole extent of the municipality?
-  Line 162: Why normalise by logarithm? Is this common practice or was a tentative normalisation procedure that ended up in good results?
- Line 173: a percentage is always between 0 and 1. So you might do the difference starting by  1 ... otherwise you need a factor 100 of scaling. I am convinced that a numerical example of FTBA would greatly help the reader.

- Line 180: The section 2.6 is quite cumbersome .Some definitions, such as "p" and "q", are given and never used in any formula or text.  To do some clustering between elements, the elements need to be compared by a distance function (which may need to respect some mathematical constraints.) If I have understood correctly, every element of your set is a series  [ $DSRp\_i, FTBA\_i$ ], with the several fixed points for DSRp  that are common for every municipality and  $FTBA\_i$ that change  accordingly (That is, a disaggregated version of figure 4). The distance is then the correlation between the set of $FTBA\_i$ of one municipality and the corresponding set  $FTBA\_j$  of another one.  If that so, please state in line 194 who is m ( number of analysed municipalities I guess) and n (the number of (equi-distant? ) sampling points in the DRSp scale, I guess).
- Formula 3: specify the upper range of any sum.
- Line 200: this kind of practical example is what makes at least the last part of 2.6 understandable.
- Line 395: does this apply to the Portugal / Southern Mediterranean area? I remind of EUCPM activation of the Czech- German border of July 27 when the FWI was not so high in the area yet several hecteareas of forest burned triggering the european activation. https://reliefweb.int/report/czechia/czech-republic-forest-fire-dg-echo-hzs-ustecky-jrc-effis-media-echo-daily-flash-26-july-2022 and https://erccportal.jrc.ec.europa.eu/ECHO-Products/Echo-Flash#/daily-flash-archive/4551
- Table 3: Nearly all the mathematical formulas need revision. (for example, "x" is a variable, not the LaTeX symbol "\times" which produces the right operator; Log(Accumulated BA) description is wrong; BNA writings are in formula format, not in text mode, and they therefore appear stretched; the same for BAF, BAS, BAA.

---

## Author Response (AR3)

The authors improved the manuscript compared to the original version. However, there is still a need for clarity, consistency, and specificity to enhance the manuscript's quality. Also, the paper still benefits from English editing to remove unnecessary words and thus be more specific and easier to read.

**Answer:** We appreciate and thank the reviewer's commentary. We follow the suggestions of the reviewer and addressed the major corrections.

The lines identified in the answers are for the track changes document, considering that the reviewer's commentaries were made identifying the lines in this file.

Abstract

• On page 2, line 33, to add the context. E.g., to support fire management. Furthermore, the authors should follow a similar in the abstract and introduction for consistency. Currently, the abstract start with the link between fire weather indices and fire behaviour, while the introduction defines the term fire regime.

**Answer:** We edited the abstract as the reviewer suggested. We also changed the Introduction, so now both starts with the link between fire weather indices and fire behaviour. The change in the Abstract regarding this subject is in line 32. The changes in the Introduction will be detailed forward in this document.

• On page 2, line 35, add the knowledge gap after Portugal. E.g., but it is still poorly understood to support fire management.

**Answer:** We add the knowledge gap after Portugal, similar with the reviewer's suggestion, in lines 33-34.

• On page 2, line 46, add the object of the paper itself before reporting the results. It is beneficial for readers to get overall the content of the article. Similarly, on page 11, line 236. E.g. This paper clarifies the effectiveness of the DSRp for estimating BA in Portugal and how vegetation influences it.

**Answer:** The object of paper itself was added as the reviewer suggested, in lines 43-34. We also added a sentence regarding this subject in the end of the Introduction, in lines 206-207.

• On page 3, line 58, remove the space in 'KEY WORDS'.

**Answer:** Corrected, in line 53.

Introduction

• On page 4, lines 61: 66, the authors should assess if that information is needed or specifically start by stating burned area as a significant component of fire regimes as in line 67 and continue providing some context on the fire weather and vegetation drivers and why this is relevant for fire management.

**Answer:** We agree with the reviewer and moved the fire regime paragraph to other part of the Introduction, in lines 114-120. Consequently, we also changed the beginning of this chapter, now in lines 62-67.

• Generally, the authors could improve the introduction by following a more logical and accessible structure for clarity. E.g., burnt area, fire weather, vegetation content, knowledge gap, research goals and a statement summarising what the paper is presenting. Furthermore, as stated in the title, the authors should be consistent with the content order in all the sections. That would make the paper easier to read.

**Answer:** We understand and appreciate the reviewer's constructive commentary and suggestions. Accordingly, we changed the Introduction structure and some sentences. We believe that this chapter is now is clearer and easier to read. The Introduction is now in lines 62-207.

Methods

• The authors reported burnt and vegetation data in the same subsections, 2.3 and subsection 2.4. is using the clusters from subsection 2.5. Thus, the authors should simplifier further the method section into a more logical and specific structure. E.g., 2.1. Study area, 2.2. Burnt area, 2.3. Fire weather, 2.4. Vegetation, 2.5. Analysing clusters of burnt area. 2.6. Analysing burnt area and fire weather relationship, 2.7. Analysing the influence of vegetation on the fire-weather relationship.

**Answer:** We agree with the reviewer and created the subsections similar as suggested. The subsection 2.1 is now in lines 210-240; 2.2 is in lines 242-264; 2.3 is in lines 265-280; 2.4 is in line 307-310; 2.5 is in lines 312-354; 2.6 is in lines 357-388 and 2.7 is in lines 390-423.

• Furthermore, the authors should assess if the subsections 2.6 and 2.7 should focus on the data analysis (e.g., from line 403 in subsection 2.4) and thus move any data preparation to the data subsection.

**Answer:** We've taken into account the reviewer's recommendations when moving the information for each subsection. We believe that the Methods section is now much more comprehensive and simpler.

• Also, they should refer to table 1 in 2.2, 2.3 and 2.4 subsections and table 2 in subsection 2.2.

**Answer:** We agree and referred the table 1 in the respective subsections 2.2 (in line 262), 2.3 (in line 271) and 2.4 (in line 310); and the table 2 in the subsection 2.2, in line 262.

• On page 20, line 403, change 'achieve the first objective' to the first objective.

**Answer:** Corrected, in line 340.

Results

• The authors should follow a similar structure for clarity and consistency. E.g., 3.1 Burnt area clusters, 3.2 Burnt area and fire weather relationship (at the national and municipal level), 3.3 Influence of vegetation on the fire-weather relationship.

**Answer:** We agree and changed the structure comparably with the reviewer's suggestion. However, we consider that the order should be different. In particular, we believe that the Burnt area clusters subsection will increase the reader's comprehension if it follows the subsection regarding the Burnt area and fire-weather relationship. Effectively, it was the analysis of that relationship (burnt area and fire-weather) at municipality scale that led us to comprehend a possible spatial clustering. Therefore, the subsection 3.1 "Burnt area and fire weather relationship (at the national and municipal level)" is in lines 426-520; the 3.2 "Burnt area clusters" is in lines 521-588; and 3.3 "Influence of vegetation on the fire-weather relationship" is in lines 589-685.

Furthermore, we moved Figure 6 and now is Figure 10, now in line 617. Consequently, the Figures 7 to 9 also changed their number in this version of the manuscript.

• Once a short name is defined in the text, the authors should consistently use it. For example, on page 33, line 587, change 'burnable area (BNA)' to BNA, as it was defined before.

**Answer:** We reviewed all acronyms in the text, including the one suggested by the reviewer. In particular, the example referred by the reviewer was also moved, now in line 592. The corrections were made in several sentences across the manuscript.

Discussion

• As for the previous sections, the discussion should follow a similar structure. E.g., 4.1 Burned area and fire weather relationship, 4.2 Influence of vegetation on the burnt area and fire weather relationship, 4.3 Considerations and implications for management.

**Answer:** We created the subsections suggested and moved several sentences. Additionally, we add two sentences in the 4.3 subsection that we consider important regarding the subsection title, in lines 799-802. The new subsection 4.1 is in lines 688-714; the 4.2 in lines 716-760; and the 4.3 in lines 762-803.

• On page 47, line 788, to re-edit for clarity. The sentence would be more appealing, stating a clear, meaningful message. The authors should do with every main message in each subject in the discussion. E.g., the main message on page 48, lines 806:807.

**Answer:** Accordingly, the two sentences were edited, now in lines 689-690 and 696-700, respectively. We carefully reviewed the Discussion section and edited some sentences.

• On page 51, line 898, change 'a very' to 'an'.

**Answer:** Corrected, in line 770.

Conclusions

• On page 54, line 952, remove 'usually'.

**Answer:** Removed, now in line 806.

• On page 54, line 953, change 'resolution' to 'resolutions.

**Answer:** The word was corrected, now in line 808.

• On page 54, lines 955:959, to simplify the message and change 'As far as we know' to 'To our knowledge.

**Answer:** We corrected the sentence, now in lines 812-813.

• On page 55, lines 963:967, change the order of the last two sentences. The implications should go to the end of the conclusions.

**Answer:** We agree with the reviewer and changed the order of the last two sentences, in lines 820-822.

Figures and tables:

• On table 1, show first burnt area metrics, then fire weather indices and finally vegetation for consistency.

**Answer:** The table 1 was changed, in line 263.

• In table 2, define the two following burn area metrics: BNAF/BNAS and TBAF/TBAS, for example, as the ratio of forest BNA and shrubland; the ratio of forest TBA and shrubland TBA. To keep consistency, edit the area of Forest in capital letters.

**Answer:** We appreciate the reviewer's suggestion and changed those burn area metrics in the Table 2, in lines 332-333.

• In figure 2, add a space between 'Nº' and 'Fire' and ':'.

• In figure 3, add spaces in the information provided in the plot area and change 'p-value' to 'p-value' for consistency.

**Answer:** We changed the Figures 2 and 3, now in lines 444 and 448, respectively.

• On figure 11 caption, line 746, add '(BA)' after 'Burnt area'. Also, in line 747, change 'Area' to 'area'.

• To add and define all the short names (e.g., 'DSR') in figure captions. E.g., on line 777, to specify 'BA'.

**Answer:** The figure 11 caption was changed, in lines 647-648. Additionally, we changed the specific example of the reviewer in a table caption (now in line 677) and also reviewed all the short names in the manuscript.

References

• On page 59, line 179:1080, double-check the reference.

**Answer:** We regret that this citation is not available online. We substituted this reference by one more updated. The new reference is in lines 979-981.

• On page 65, line 1226, remove '(Table 1),'.

**Answer:** Corrected, in line 1049.

• On page 66, line 1247, remove ','.

**Answer:** The reference was corrected, specifically in line 1070.

---

## Author Response (AR4)

Dear Editor,

We are thankful to the reviewer and the editor for your constructive commentaries and suggestions.

According to your suggestions, we added several sentences to explain and highlight the novelty of this work, both in the Abstract and in the Conclusions, and the importance of our results for increasing the knowledge in fire science.

All of the reviewer's suggestions, comments, and questions deserved our best attention. Please check our responses and comments to the reviewer in the file "Answer to reviewer". Finally, we took the opportunity to proofread the entire manuscript and correct a small number of typos and grammatical errors.

We firmly believe that the novelty and importance of our results for the improvement of scientific knowledge are now explained much more clearly and that all the reviewer's observations, queries, suggestions and comments have been properly addressed. Thus, we hope that the manuscript can be considered acceptable for publication.

Best regards,

Tomás Calheiros (on behalf of the Authors).

Anonymous referee #4

Summary: Overall, this manuscript presents sound scientific ability and strong technical analytical skills, however, I have some significant doubts about the contribution to new science and the bridging of knowledge gaps to the field. Not only are the differences between this study and previous research not strongly argued but the lack of further contextual information about fire behaviour is an issue. The largest is the lack of consideration of local scale conditions on relatively small-scale fires ~100ha. This study uses regional scale weather inputs to assess potentially local scale drivers and influences. There is also a lack of understanding fuel characteristics that drive fire behaviour and their relationship to occurrence. The authors make an effort to address purpose of this research (through its relevance to Portugal) but it fails to outline the significance of its work in relation to other similar fire prone regions internationally and there is a lack of reference material to the broader fire science community, to which have produced similar work (conceptually). There are many missing definitions and structural issues with this manuscript, I'd strongly suggest that the authors thoroughly revise the manuscript to improve the communication of the aims and objectives. The conclusions are also not strongly supported nor adequately explained. I'd also suggest that there needs to be some further work to highlight the novel nature of the work, particularly when LULC and fire occurrence is so broadly researched internationally.

**Answer:** We thank and appreciate all the constructive reviewer's comments and suggestions, that aimed to increase substantially the clarity, quality and suitability of this manuscript for publication. We modified

the document accordingly. In particular, regarding the objectives, conclusions and novelty of the work, we changed and add several sentences in the Abstract (lines 35-39 and 47-54 of the track-changes file), Introduction (lines 160-163 and 168-173), Discussion (lines 698-703) and in the Conclusion (lines 719-727).

We want to refer that all the lines indicated in this document are related for the track-changes file.

Since the Summary is mostly divided into the Major comments, we answer those comments in the following lines. In addition, the commentaries in the PDF file are also answered or explained in the next lines.

Major Comments:

Line 40: The DSR is actually never properly addressed in terms of its composition and its strength/weaknesses versus other fire weather indices. There is a lack of depth when considering other potential metrics as well as almost no discussion about how these are produced or applied.

**Answer:** The manuscript already includes information on the DSR composition and how is produced and applied in sections 2.3 and 2.5, and also in Table 1 and Table 2. However, we agree with the reviewer that DSR was not properly addressed and compared with other indices. Consequently, we added some sentences to clarify these subjects and to explain that the DSR is widely used in Portugal, both for research and operational purposes, in lines 95-97. We also inserted a few more references to validate our statements.

Line 60: The understanding of vegetation and its role in driving large scale fires is poorly discussed. There is no reference to grasslands or ephemeral grasses, as well as many other prominent vegetation types. Further to this, I'd be certain that on the 100ha scale that Eucalypts behave very differently than other native vegetation. It is also mentioned that Eucalypts are not a significant driver in the change in fire regime, however it makes up the largest percentage of fuels in Portugal. This needs to be explained better.

**Answer:** In general, we agree with the reviewer's observations. However, it is important to emphasize that:

1- In particular, grasslands or ephemeral grasslands are not an important vegetation type in Portugal, covering only 7% of the territory;
2- Grasslands are present mostly in the south of the country (Alentejo), a region that is almost unaffected by wildfires;
3- Our purpose in this work was not to study the fire proneness or the fire behavior in different types of vegetation. Additionally, we only aimed to analyze burnt areas in five major land use types, including forest, but not any shrub or forest type, including Eucalypts. We added a phrase regarding this subject in lines 258-259.

Line 90: I'm not convinced about the gaps in knowledge. I don't think there has been enough effort to explain the differences between what is proposed and previous studies.

**Answer:** We agree with the reviewer that the gaps in knowledge are not properly developed in this part of the Introduction. Therefore, we modified the manuscript to clarify the gaps in knowledge, precisely in lines 160-163.

Line 95: There has been almost no effort to discuss vegetation cover and what metric will be used to assess it.

**Answer:** The vegetation data used in the study is only presented in Section 2.4, in the Data and Methodology section. We discuss the vegetation cover and its influence on the relationship between fire and weather in Section 3.3, based on several metrics including Burnable area (BNA), Forest/Shrubland burnable area (BNAF/BNAS), Forest/Shrubland total burnt area (TBAF/TBAS) and Burnt area (BA) in three major land use types. All the metrics are defined in section 2.4 "Vegetation and land use data" and in Table 2.

Nevertheless, we understand the reviewer's doubts and, consequently, we add more information about the data to clarify this subject, together with a new reference, in Section 2.4, in lines 254-259.

Line 135: Is 1200UTC the most suitable reference time for Portuguese fires? Peak conditions are typically around 1400 local time? This has not been discussed at all.

**Answer:** We agree with the reviewer that 1200UTC is not the hour of the most fire-prone conditions in Portugal during summer. However, according to Van Wagner and Pickett (1975), the indices of the CFFWIS should be computed with meteorological daily data registered at noon, namely air temperature and relative humidity, wind speed, and accumulated total precipitation, as explained in section 2.3 of the manuscript.

Line 140: 9km is big when considering 100ha fires. Your weather data is roughly 9 times more course than the ignition data? I know this study is generally considering regional drivers, however, this is not robustly considered or discussed. Perhaps but is probably not a good dataset for considering regional scale fire weather and ignitions. Also, you have listed just two "worldwide" studies.

**Answer:** We understand the reviewer's worries regarding the weather and fire data resolution. However, it is important to clarify that:

1. We did not study wildfires with 100 ha, but wildfires with the burnt area above 100 ha. The burnt area median of the 2016 wildfires is 303 ha.
2. Climate reanalysis combines past observations with models to generate consistent time series of multiple climate variables. This is why reanalysis is among the most-used datasets in the geophysical sciences.
3. ERA5 is the latest climate reanalysis produced by ECMWF.
4. The ERA5-Land is the database with the highest spatial resolution, providing all the meteorological data needed to compute the fire weather indices for Portugal and for the study

period. This reanalysis dataset is used in weather and climate research and is frequently used in fire weather studies.

5. Observed datasets with time series of meteorological variables measured in weather stations have much lower resolution. Downscaling the ERA5-Land or another dataset to the same resolution of the wildfires dataset would add considerable uncertainty, due for instance the local impacts of topography, with limited added value for this work.

6. Meteorological conditions are similar in relatively large areas, at municipal scale or higher, considerably much higher that burnt areas, especially on days when large wildfires occur.

Therefore, we modified some sentences to clarify the text and added more references, in lines 245-250.

**Comments of the reviewer provided in the PDF file and respective answer:**

Line 2: Why is this important and what about other indicies?

**Answer:** This subject is detailed in the answer to the Major Comments of Line 40 (in previous lines of this document).

Line 3: Is the DSRp only used at the macroscale? What's smaller scale?

**Answer:** We regret this lack of attention from us. We modified the sentence to "spatial smaller scale" to increase clarity.

Line 4: Now there is a reference to "large" burnt areas?

**Answer:** We corrected and clarified the sentence.

Line 5: poorly worded. rephrase

**Answer:** We agree with the reviewer and rephrased this sentence.

Line 6: through what metric?

**Answer:** The abstract has a word limit and we consider that the space needed to explain this part is better placed in Section 2.4. Please also see the answer for the Major Comment of line 95, in the previous lines of this document.

Line 7: Portuguese land management departments

**Answer:** We agree that the reviewer's suggestion could increase the clarity and changed the line.

Line 9: 100ha isn't particularly large

**Answer:** The concept of large or extreme wildfires is statistical and depends on the sample and the study region. The objective of this study is not to establish the 100 ha limit to define a large wildfire. The objective is to find a limit that allows us to consider the minimum number of wildfires that explains a large part of the burnt area. Additionally, we explained in the manuscript and justified this 100 ha limit in lines 216-223.

Line 10: What's an extreme day for Portugal?

**Answer:** An extreme event is also a statistical concept. Usually, an extreme event is defined for values above a high percentile, e. g. 90th percentile. In Portugal, extreme fire weather days correspond to days when the DSR is greater than the 90th DSR Percentile. This limit was previously used by several authors (please see lines 97-102). Additionally, as stated in lines 300-303 of the manuscript: "We considered the correspondent 80% and 90% of FTBA as sufficient to classify DSRp as the extreme threshold, justified by the results of Pereira et al., (2005), which showed that 80% of TBA occurs in 10% of summer days."

Our results confirm that this limit is adequate, as said in lines 603-604: "According to our results, only 6% of the TBA occurs with DSRp<80 and 12% of TBA are registered in wildfires with DSRp<90".

Line 13-14: Is this not just due to the potential differences in rate of spreads between the two vegetation types?

**Answer:** We agree with the reviewer and recognize his pertinent observation. Effectively, meteorological conditions influence the Rate Of Spread (ROS). With a lower DSR, the ROS in shrublands should be higher than in the forest species, while a higher DSR allows higher ROS in forests. Nevertheless, we did not study the rate of spread, only analyzed the final area burned by the wildfire. We consider that is advisable not to make any consideration about ROS because we did not study this fire behavior characteristic.

Line 17: Semi-arid and arid landscapes?

**Answer:** The Mediterranean Climate corresponds to the Csa and Csb type of climate according to the Koppen climate classification. We add this information in the manuscript, in line 61.

Lines 19-20: Name the Koeppen climate classification or perhaps just use that definition

**Answer:** We accept the reviewer's suggestion and add the Koppen climate classification (Csa and Csb).

Lines 21: increasingly evident

**Answer:** We accept the reviewer's suggestion and modified the sentence.

---

## Author Response (AR5)

Dear Editor,

We are very thankful to the editor for your decision to accept our manuscript for publication and for the opportunity to answer another reviewer. Indeed, this reviewer made very pertinent questions, comments and suggestions, some already considered in the previous review, but others new that helped to clarify some parts of the manuscript.

Best regards,

Tomás Calheiros (on behalf of the Authors).

Review for "Drivers of extreme burnt area in Portugal: fire weather and vegetation" of T.

Calheiros et al.

In this paper, the relation between Daily Severity Rating percentile (DSRp) and the total

burned area (BA) in Portugal is studied, with the aim of understanding its smaller scale

(municipal scale) behaviour. The Authors tried to 1) assess if the performance of 90th DSRp

(DSR90p) threshold in BA prediction in mainland Portugal; 2) identify and characterise regional

variations of the DSRp threshold that justifies the majority of Burned Area ; and 3) analyse if

the DSRp spatial variability could be explained with broad classification of land cover (forested

vs agricultural vs shrubbed).

As a dataset, weather reanalysis data from ERA5-Land as well as wildfire and land use data

from official Portuguese authorities for an extended summer period (15th May to 31st October)

from 2001 to 2019 were used.

The treated topic is exceptionally relevant, since fire weather indices can and should be coupled

by info on vegetation for optimal wildfire management procedures.

However, the paper should be refined in some parts before being considered for publication.

- Line 43 Define DSR (or at least specify that is a simple reformulation of FWI). This can be done here or at line 135.
  **Answer:** Another reviewer also pointed out this issue and, consequently, we added information to clarify this aspect.

- Line 43 The reader needs to understand what a DSR percentile is. In order to get a

threshold based on percentiles, we need a set of elements to be sorted in ascending order. On which set were the percentile classes defined? This needs clarification.
**Answer:** In the last revision of the version of the manuscript we clarify how the DSRp is computed including the sorting and the dataset used in Section 2.5.

- Line 115: how the burnt area dataset is derived? Polygons retrieved from ground assessments? Satellite?
  **Answer:** We agree with the reviewer that this information is needed. We modified the text to explain that burnt area polygons are derived from satellite data.

- Line 115: of course, the threshold of 100 ha applies to European fire regime and not to, e.g., North American one.. Maybe this thought can be added in the text.
  **Answer:** Another reviewer also discussed this subject. The concept of large or extreme wildfires is statistical and depends on the sample and the study region. The objective of this study is not to establish the 100 ha limit to define a large wildfire. The objective is to find a limit that allows us to consider the minimum number of wildfires that explains a large part of the burnt area. We explained this in the manuscript and justified this 100 ha limit, in the previous review.

- Line 130 Why is air temperature and not air humidity the driver for fires?
  **Answer:** A previous study cited in the manuscript (Amraoui et al., 2015) shows that while the summer season peak of fire incidence is more dependent on air temperature higher values, the winter peak is much more dependent on low values of air humidity because in this season the air temperature is usually the lowest. This is also already explained in the manuscript.

- Line 145 How was the original classification of COS2018? How was the aggregation performed?
  **Answer:** We agree with the reviewer and clarify this in the new revised manuscript (in Section 2.4).

- Line 157 : "was allocated to this administrative unit". What unit are they talking about? The sentence can be reformulated.
  **Answer.:** We agree with the reviewer that the sentence was not clear. Consequently, we modified it to clarify the administrative unit.

- Line 160: Still not clear what BA percentages is.
  **Answer.:** We agree with the reviewer and clarified the text.

- Line 163: So for each fire event at municipal scale, the maximum DSR is selected in the days of the event and the whole extent of the municipality?
  **Answer.:** Of course, we could have used other statistics (e.g. the mean). However, it is important to underline two aspects. The first is that the weather conditions on each day are very similar in relatively large regions, large than the municipalities. The second is that we are interested in identifying extreme fire weather associated with large wildfires. We already mentioned in the manuscript that "The selection of the maximum value of DSR to associate with wildfires is justified

by the low spatial variability of the DSR, the small size of administrative units and the native reanalysis data resolution (C3S, 2020)".

- Line 162: Why normalise by logarithm? Is this common practice or was a tentative normalisation procedure that ended up in good results?
**Answer.:** The logarithm (of burnt area) should be computed for several reasons. One is when the relationship between two variables is not linear but exponential. Another is when you want to apply a method that can only be used on normally distributed data and your data does not fulfil this requirement. All these situations apply in this case. In addition, we note that the accumulated BA (after sorting the BA time series from lower to higher DSR values), varies exponentially with the DSR percentile. It is also worth mentioning that the use of the logarithm is a common practice in burnt area distribution studies.

- Line 173: a percentage is always between 0 and 1. So you might do the difference starting by 1 … otherwise you need a factor 100 of scaling. I am convinced that a numerical example of FTBA would greatly help the reader.
**Answer.:** The reviewer presents an important question. Indeed, we started this analysis precisely following the reviewer's suggestion. However, after a long discussion about how to present these results, we decide that the adopted one was the best and simple way for the readers to visualize and understand the DSR percentile limit that corresponds, for example, to 90% of burnt area. But we agree that it is a debatable decision.

- Line 180: The section 2.6 is quite cumbersome .Some definitions, such as "p" and "q", are given and never used in any formula or text. To do some clustering between elements, the elements need to be compared by a distance function (which may need to respect some mathematical constraints.) If I have understood correctly, every element of your set is a series [ DSRp_i,FTBA_i ], with the several fixed points for DSRp that are common for every municipality and FTBA_i that change accordingly (That is, a disaggregated version of figure 4). The distance is then the correlation between the set of FTBA_i of one municipality and the corresponding set FTBA_j of another one. If that so, please state in line 194 who is m ( number of analysed municipalities I guess) and n (the number of (equi-distant? ) sampling points in the DRSp scale, I guess).
**Answer.:** We agree with the reviewer that "p" and "q" should not appear in this text, because they are not necessary in our case. We corrected the manuscript. We thank the other reviewer's suggestions and added that information, in lines 326-332 of the new version of the manuscript (in the track changes version).

- Formula 3: specify the upper range of any sum.
**Answer.:** We appreciate the reviewer's suggestion but we believe that is sufficient in the formula the letter $j$. These formulas were obtained in Matlab software and also in the Mathworks website, as referred in the manuscript.

- Line 200: this kind of practical example is what makes at least the last part of 2.6 understandable.

**Answer.:** We thank the reviewer's comment.

- Line 395: does this apply to the Portugal / Southern Mediterranean area? I remind of EUCPM activation of the Czech- German border of July 27 when the FWI was not so high in the area yet several hecteareas of forest burned triggering the european activation. https://reliefweb.int/report/czechia/czech-republic-forest-fire-dg-echo-hzs-ustecky-jrc-effis-media-echo-daily-flash-26-july-2022 and https://erccportal.jrc.ec.europa.eu/ECHO-Products/Echo-Flash#/daily-flash-archive/4551
  **Answer:** Our results clearly show that large wildfires can occur with relatively low DSR. However, the largest and most extreme wildfires only occur under extreme fire weather and DSRp. In addition, as mentioned in lines 395-397, our results also show that "forests tend to burn only under extreme DSR values, typically caused by simultaneous drought and heatwave, while shrublands (and also agricultural areas) can burn with lower DSRp". However, these results were obtained for Portugal. We suppose that similar results can be obtained for the Mediterranean Basin. Nevertheless, this relationship must be detailed checked in other European regions with a different type of climate.

- Table 3: Nearly all the mathematical formulas need revision. (for example, "x" is a variable, not the LaTeX symbol "\times" which produces the right operator; Log(Accumulated BA) description is wrong; BNA writings are in formula format, not in text mode, and they therefore appear stretched; the same for BAF, BAS, BAA.
  **Answer:** We accept the reviewer's suggestions and modified the LaTeX program, replacing all the "x" with "\times". Additionally, we altered the formula writings to text mode in BNA, BAF, BAS and BAA formulas, also in the LaTeX program.